# Engineered *E. coli* Nissle 1917 for the delivery of matrix-tethered therapeutic domains to the gut

Pichet Praveschotinunt [1,2], Anna M. Duraj-Thatte[1,2], Ilia Gelfat [1,2], Franziska Bahl[1,3], David B. Chou [1,4] & Neel S. Joshi [1,2]*

Mucosal healing plays a critical role in combatting the effects of inflammatory bowel disease, fistulae and ulcers. While most treatments for such diseases focus on systemically delivered anti-inflammatory drugs, often leading to detrimental side effects, mucosal healing agents that target the gut epithelium are underexplored. We genetically engineer *Escherichia coli* Nissle 1917 (EcN) to create fibrous matrices that promote gut epithelial integrity in situ. These matrices consist of curli nanofibers displaying trefoil factors (TFFs), known to promote intestinal barrier function and epithelial restitution. We confirm that engineered EcN can secrete the curli-fused TFFs in vitro and in vivo, and is non-pathogenic. We observe enhanced protective effects of engineered EcN against dextran sodium sulfate-induced colitis in mice, associated with mucosal healing and immunomodulation. This work lays a foundation for the development of a platform in which the in situ production of therapeutic protein matrices from beneficial bacteria can be exploited.

[1] Wyss Institute for Biologically Inspired Engineering, Harvard University, Boston, MA, USA. [2] John A. Paulson School of Engineering and Applied Sciences, Harvard University, Cambridge, MA, USA. [3] Faculty of Biology, Albert Ludwigs University of Freiburg, Freiburg im Breisgau, Germany. [4] Department of Pathology, Massachusetts General Hospital, Boston, MA, USA. *email: neel.joshi@wyss.harvard.edu

nflammatory bowel disease (IBD) describes a group of auto-immune diseases that cause chronic inflammation of the small and large intestine. This group of diseases affects about 3 million adults in the U.S.[1]. A complete picture of IBD etiology remains a subject of intense research and debate, and the development of the disease has been linked to multiple factors, including dysregulated immune responses, genetic predisposition, and an altered balance of microbiota (i.e., dysbiosis)[2]. However, the complex interplay between these factors has greatly hindered the development of effective therapies. Conventional IBD treatment relies on pharmacological interventions that scale with the severity of the disease, starting with aminosalicylates and antibiotics, and proceeding to corticosteroids and immunosuppressants, with the goal of dampening inflammation by influencing host biology or by decreasing the chance of bacterial infection of mucosal wounds. The consequences of disease flare-ups can be severe and mount over time, leading to 23–45% of ulcerative colitis patients and 75% of Crohn's disease patients requiring surgical removal of portions of their gastrointestinal (GI) tract at some point in their lives[3]. Therefore, the ability to induce deep remission and sustain it indefinitely is the long-term goal of IBD treatments.

Some early successes with therapies that target tumor necrosis factor (TNF) initially indicated a promising future for biologics in the treatment of severe cases of IBD. However, progress in the development of new therapeutics has been slow, with several notable clinical failures arising from drug candidates with strong results in small animal models[4]. A common theme in these failures is that the efficacy for a given treatment varies according to patient sub-population or environmental factors (e.g., diet, social behaviors). Given the heterogeneity in disease etiology, it is likely that multi-pronged and perhaps patient-specific management strategies will be necessary to achieve effective clinical outcomes[5].

During IBD disease flare-ups, the barrier formed by the GI epithelium is disrupted, exposing the gut lining to potentially injurious agents like bacteria and their products or extreme pH[6]. This is true for other chronic inflammatory symptoms like ulcers or fistula as well. Therefore, the speedy restoration of mucosal healing and epithelial integrity is essential for treating such symptoms. The epithelial mucosa heal through restitution and regeneration processes. Complete regeneration is a slower process that relies on stem cell proliferation and differentiation. In contrast, restitution can occur within hours after injury and relies on the migration of epithelial cells from the surrounding area into the wound site. This process can restore mucosal continuity to the gut lining and protect it from bacteria and foreign antigens, and fluid and electrolyte losses, which prevent further inflammatory processes[7]. Although epithelial restitution is essential in protecting the GI tract during insult, it is difficult to monitor as an outcome directly in clinical studies[8]. Most therapeutics for IBD and ulcers focus on modulating inflammatory pathways, leaving room for therapeutic advancement in mucosal healing.

An emerging area of IBD research deals with gut microbes. Although it remains unclear whether dysbiosis can incite disease, it is clear that global reductions in gut microbiome diversity are correlated with IBD severity[9]. Furthermore, it is also clear that many microbiota can exacerbate IBD-associated inflammation via compromised epithelial barrier function[10]. These factors have led to the exploration of living bacteria as therapeutic entities that can be delivered orally (i.e., bugs as drugs). Several naturally occurring commensal and beneficial strains have been explored as therapeutics, with limited success mostly stemming from their low potency and inability to persist in the GI tract[11,12]. Genetically engineered microbes have also been explored, mostly as a means to secrete biologic drugs (e.g., interleukin (IL)-10, anti-tumor necrosis factor (TNF)) locally in the colon[13–15]. Many such

efforts have also shown high efficacy in animal models but have yet to yield clinical successes, in part because of difficulties in achieving and maintaining sufficiently high concentrations of the therapeutic molecule at the site of disease. Indeed, concerns have been raised about the compatibility of this strategy with immunomodulatory biologics, since the mucosal epithelial barrier hinders their trafficking to their target cells in the lamina propria[16,17]. Nevertheless, the promise of effective treatments that can be produced cheaply, delivered orally, and minimize systemic side effects has continued to fuel interest in microbes as therapeutics.

Here, we present an alternative approach to engineered microbial therapies to promote mucosal healing. Instead of secreting soluble therapeutic proteins, we programmed bacteria to assemble a multivalent material decorated with anti-inflammatory domains in the gut. The displayed domains are designed to target the material to the mucosal layer of the epithelium and promote host processes that reinforce epithelial barrier function (Fig. 1). The bacterially produced scaffold for the living material is based on curli fibers, a common proteinaceous component of bacterial extracellular matrices. Hence, we refer to our approach as probiotic-associated therapeutic curli hybrids (PATCH). We demonstrate that PATCH is capable of ameliorating inflammation caused by dextran sodium sulfate (DSS)-induced colitis in a mouse model.

## Results

**Probiotic-associated therapeutic curli hybrids (PATCH).** We used *E. coli* Nissle 1917 (EcN) as our cellular chassis for PATCH. EcN is well-studied, has a long track record of safety in humans, and is a popular starting point for engineered therapeutic microbe efforts because of its compatibility with canonical genetic engineering techniques for bacteria[18]. In addition to its use as an over-the-counter supplement for general GI disorders, EcN has also been evaluated in comparison to mesalazine for maintaining remission in ulcerative colitis in randomized control trials[19]. While EcN led to some favorable outcomes, overall efficacy was low and relapse rates were high, impeding its use as a first-line treatment for IBD[19,20]. Like other Enterobacteriaceae, EcN resides mostly in the colon, where it colocalizes with areas affected by many types of IBD[21]. Moreover, facultative anaerobes like EcN are known to thrive in the highly oxidative environment of the inflamed GI tract[22], making EcN an ideal starting point for our engineering efforts.

We chose the trefoil factor (TFF) family of human cytokines as our bioactive domain for display on curli fibers. TFFs are small, 7–12 kDa proteins secreted by mucus-producing cells in the GI tract and other mucosal surfaces, primarily to promote epithelial restitution[7]. TFFs also reportedly have tumor suppressing, apoptosis blockading, and barrier function augmenting bioactivity, though the precise mechanisms for these effects are still not well understood[7,23]. TFFs have been explored for IBD treatment, but oral delivery did not yield therapeutic outcomes, as they were found to adhere too strongly to the mucus layer of the small intestine[15]. We sought to overcome this by tethering them to the curli fiber matrix after local production in the ileum, cecum, and colon.

**Secretion and self-assembly of EcN-derived TFF-fused curli.** In order to implement the PATCH system, we created plasmid-based genetic constructs encoding for the self-assembling monomer unit of curli fibers (CsgA) fused to each of the three TFFs (TFF1-3). The TFFs were appended to the C-terminus of CsgA via a flexible glycine-serine linker containing an internal 6xHIS tag in a manner that we have previously shown to not interfere with extracellular

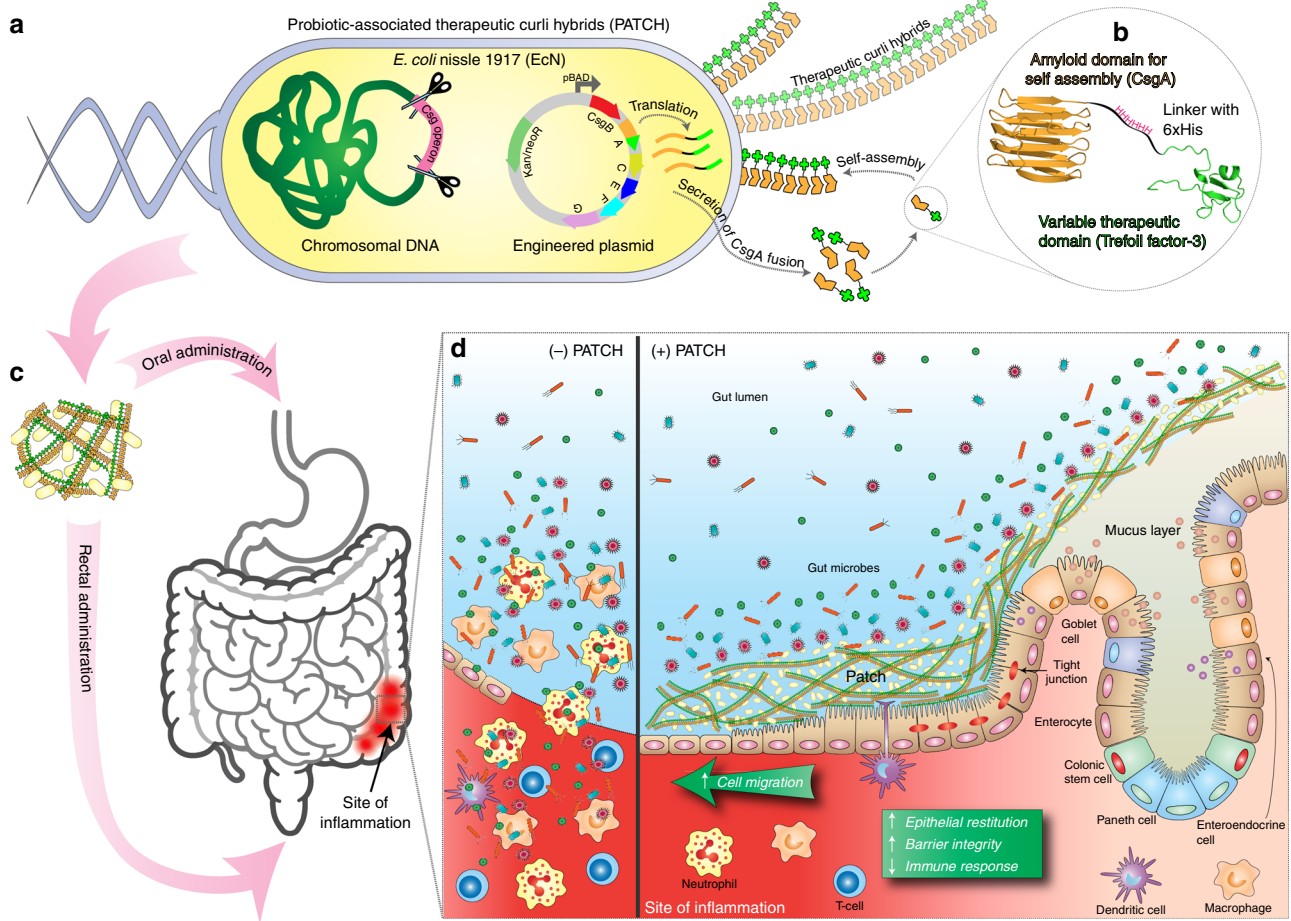

**Fig. 1 Probiotic-associated therapeutic curli hybrids (PATCH). a** Schematic overview of engineered curli production. Genetically engineered *E. coli* Nissle 1917 (EcN) with csg (curli) operon deletion (PBP8 strain) containing plasmids encoding a synthetic curli operon capable of producing chimeric CsgA proteins (yellow chevrons with appended bright green domains), which are secreted and self-assembled extracellularly into therapeutic curli hybrid fibers. **b** CsgA (yellow), the main proteinaceous component of the *E. coli* biofilm matrix, was genetically fused to a therapeutic domain—in this case, TFF3 (PDB ID: 19ET, bright green), which is a cytokine secreted by mucus-producing cells. The flexible linker (black) includes a 6xHis tag for detection purposes. **c** Engineered bacteria are produced in bulk before delivery to the subject via oral or rectal routes. A site of colonic inflammation is highlighted in red. **d** Interaction of PATCH and the colonic mucosa. Inflammatory lesions in IBD result in loss of colonic crypt structure, damage to epithelial tissue, and compromised barrier integrity (left panel, (−) PATCH). The resulting invasion of luminal contents and recruitment of immune cells to the site exacerbates the local inflammation. The application of PATCH (right panel, (+) PATCH) reinforces barrier function, promotes epithelial restitution, and dampens inflammatory signaling to ameliorate IBD activity.

secretion and self-assembly[24]. The library of plasmids was designed such that each gene encoding a CsgA-TFF fusion was co-transcribed with the other genes necessary for effective curli secretion and assembly (*csgB, csgC, csgE, csgF, csgG*)[25–27]. Altogether, these formed a synthetic curli operon that was placed under the control of an inducible promoter (P$_{BAD}$) in a pBbB8k plasmid backbone bearing a kanamycin selection marker (Fig. 1). The inclusion of the other genes of the curli operon was necessary to increase secretion efficiency, because the curli genes in the EcN chromosome are downregulated at physiological temperature and osmolarity[28]. Nevertheless, we also employed an EcN mutant in which all of the chromosomal curli genes were deleted (EcN Δ*csgBACDEFG::Cm^R*, a.k.a PBP8) in order to preclude the possibility of curli fiber expression from native genes confounding our experimental results[29].

In order to confirm that curli fibers decorated with TFFs could be produced by EcN, as they can in laboratory strains of *E. coli*[24], we transformed EcN with the panel of synthetic curli plasmid constructs, in addition to a vector encoding GFP in place of the curli genes as a negative control. The transformed cells were cultured at 37 °C in high-osmolarity media to mimic physiological conditions and induced with L-(+)-arabinose. A quantitative Congo Red-binding (CR) assay, normally used for curli fiber detection[24,30], indicated that wild-type CsgA and all three CsgA-TFF fusions could be expressed and assembled into curli fibers under these conditions, while EcN with the GFP-expressing control vector showed no CR binding (Fig. 2a). Extracellular assembly was further confirmed using whole-cell enzyme-linked immunosorbent assay (ELISA) assays probing for the 6xHIS tag (Fig. 2b). Scanning electron microscopy of the samples confirmed that recombinant wild-type CsgA and all the CsgA-TFF fusions assembled into nanofibrous structures resembling native curli fiber in appearance (Fig. 2c–g). Curli production experiments performed with PBP8 led to similar trends. In some cases, basal expression of the *csgA* genes was observed without induction (Supplementary Fig. 1A, B). We have also confirmed the presence of the displayed TFF3 using a similar whole-cell ELISA assay (Fig. 2h).

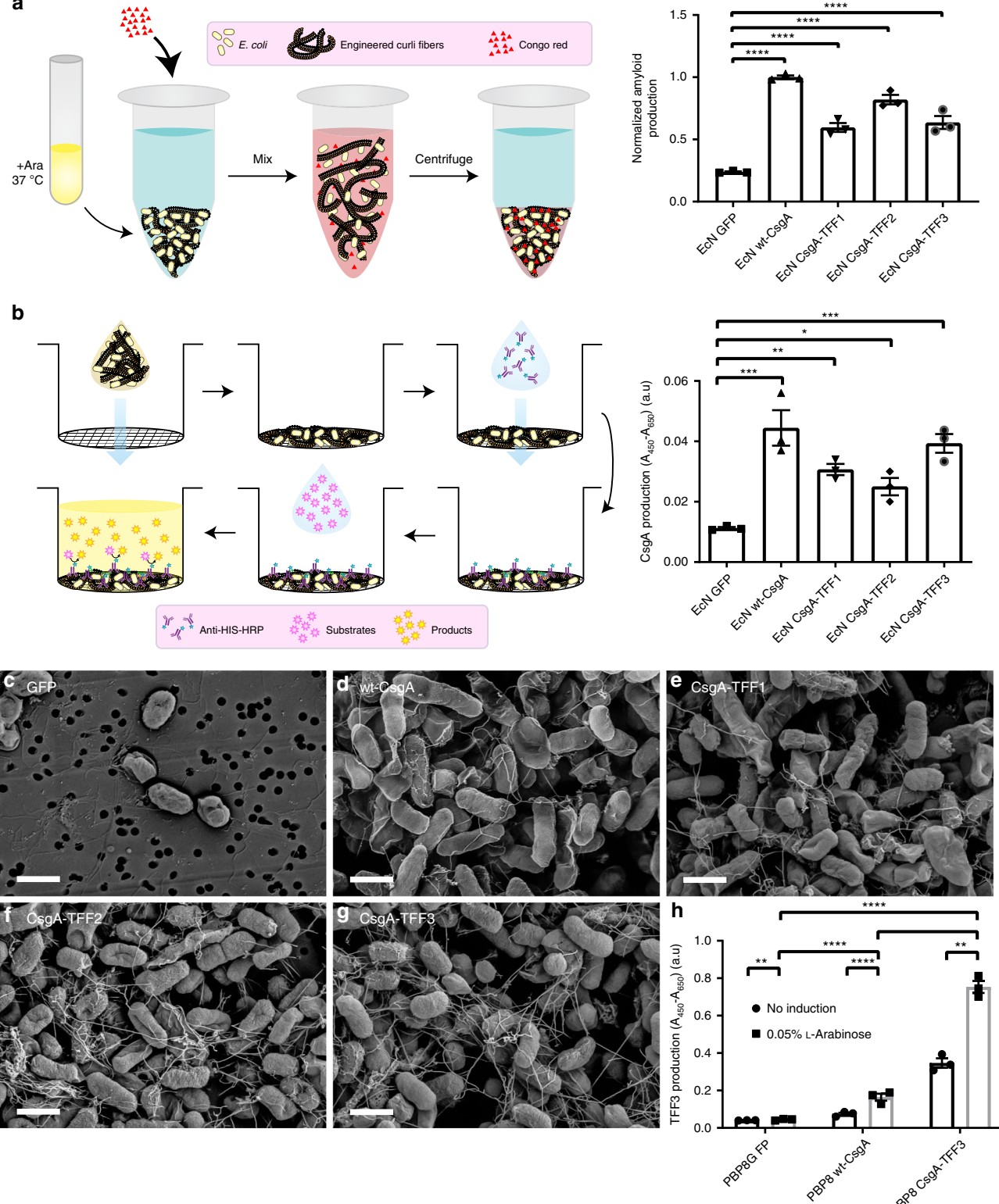

**Fig. 2 Production of curli fiber variants from engineered EcN. a** Schematic of quantitative Congo Red (CR)-binding assay (Yellow ovals = *E. coli*, Orange and green lines = engineered curli fibers, Red triangles = Congo Red) (left). Normalized amyloid production of each EcN variant, as measured by CR-binding assay (right), after induction with arabinose (Ara) at 37 °C in LB media. **b** Schematic of whole-cell filtration ELISA for the monitoring of modified curli production from bacterial culture (Purple Y shapes with blue dots = Anti-HIS-HRP, Pink stars = HRP Substrates, Yellow stars = HRP products) (left). Relative CsgA production between EcN variants, derived from anti-6xHis antibody-based detection. **c–g** Scanning electron micrographs of EcN transformed with plasmids encoding various proteins: **c** GFP **d** wt-CsgA **e** CsgA-TFF1 **f** CsgA-TFF2 **g** CsgA-TFF3 (scale bar = 1 μm). **h** Relative TFF3 production (using anti-TFF3 antibody) of induced and non-induced PBP8 library. Data are represented as mean ± standard error of the mean (SEM). (ns) $p > 0.05$, $*p \leq 0.05$, $**p \leq 0.01$, $***p \leq 0.001$, $****p \leq 0.0001$, one-way ANOVA followed by Dunnett's test. Source data are provided as a Source Data file.

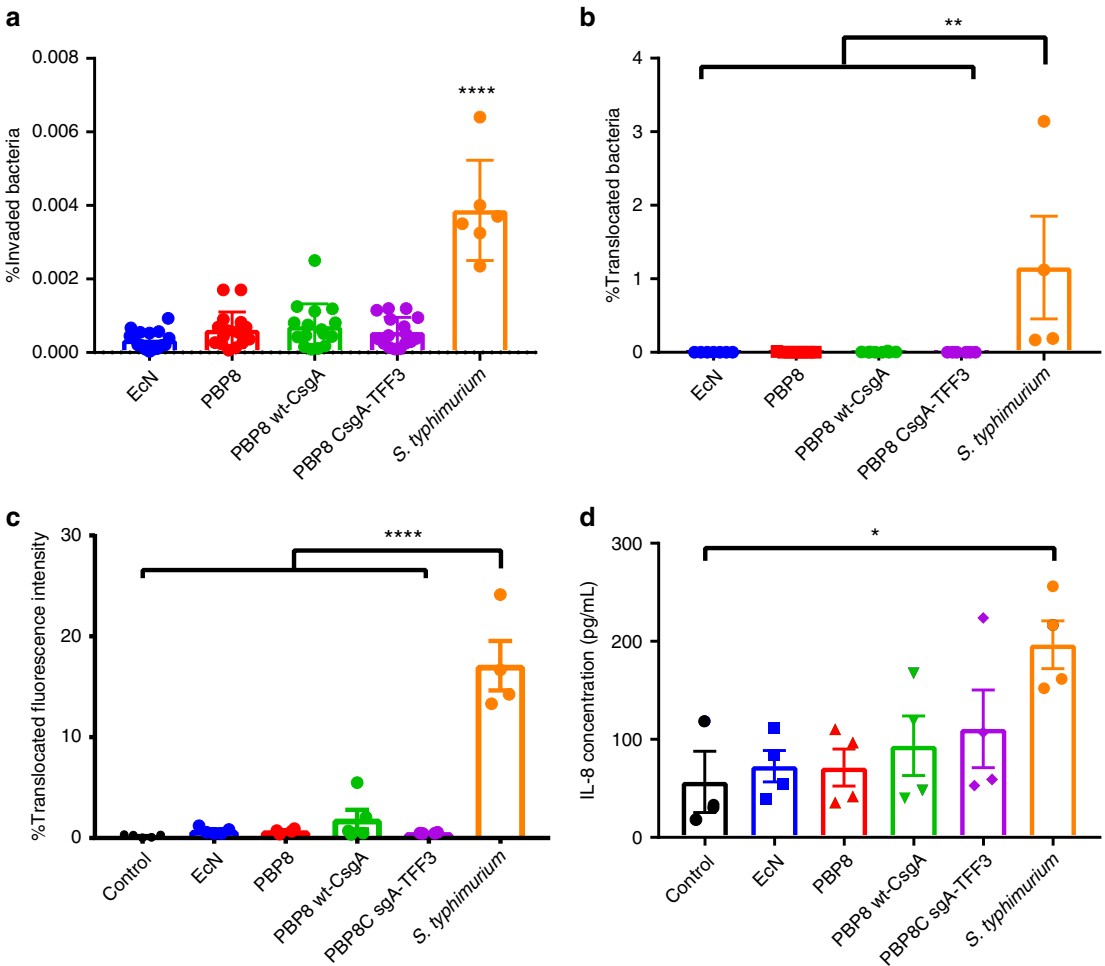

**Fig. 3 Effects of curli fiber expression on EcN pathogenicity. a** Percent of bacteria that invaded a monolayer of Caco-2 after 2 h of co-incubation with EcN, PBP8 variants, and *S. typhimurium*. **b** Bacterial translocation to the basolateral compartment of polarized Caco-2 cells exposed to bacterial library for 5 h. **c** Epithelial permeability of polarized Caco-2 24 h post-infection, quantified via FITC-dextran (MW 3000–5000) translocation. **d** IL-8 secretion from the basolateral compartment of polarized Caco-2 cells 24 h post-infection. Data are represented as mean ± SEM. (ns) $p > 0.05$, *$p \leq 0.05$, **$p \leq 0.01$, ***$p \leq$ 0.001, ****$p \leq 0.0001$, one-way ANOVA followed by Tukey's test. Source data are provided as a Source Data file.

**In vitro non-pathogenicity of modified EcN**. We had previously demonstrated that the CsgA-TFF3 fusion, when produced by a laboratory strain of *E. coli*, could bind to mucins and promote mammalian cell migration in an in vitro model with a human colorectal adenocarcinoma cell line (Caco-2)[24]. Before proceeding to in vivo studies, we wanted to confirm that modified curli fiber overproduction did not induce a pathogenic phenotype in PBP8. Therefore, we performed invasion and barrier function assays on Caco-2 cells grown to confluency in transwells. None of the EcN-derived strains (i.e., PBP8—curli genes deleted, PBP8 wt-CsgA—expressing the wild-type CsgA sequence, PBP8 CsgA-TFF3—expressing the CsgA-TFF3 fusion) exhibited increased invasiveness into polarized Caco-2 monolayers when compared to unmodified EcN (Fig. 3). Invasion for Caco-2 monolayers was low across all groups in comparison to a positive control, *Salmonella typhimurium* (SL 1344) (Fig. 3a). Similarly, in a translocation assay in which bacteria were collected from the basolateral chamber of the transwell[31], we observed essentially no translocation of any of the EcN-derived strains (Fig. 3b). We also monitored barrier function in the transwells as a function of bacterial strain. Transepithelial electrical resistance (TEER) measurements showed lower reductions to TEER values for all of the EcN-derived strains compared to *S. typhimurium*—40–50% vs. 70%, respectively (Supplementary Fig. 2). We also tested

barrier function in vitro via the translocation of fluorescently labeled dextrans by adding them to the apical chamber of the transwell after 24 h of incubation with the bacteria[31,32]. We observed almost no translocation for any of the EcN-derived strains, while the positive control (*S. typhimurium*) led to significant translocation of the polymer to the basolateral chamber (Fig. 3c). While Caco-2 cells are a crude mimic of the mucosal epithelium, they are known to respond to exposure to pathogenic bacteria in predictable ways that include the activation of pro-inflammatory signaling cascades and release of cytokines such as IL-8[31,33–36]. We monitored the response of polarized Caco-2 cells to apical bacterial exposure for 24 h and found that EcN and the PBP8 variants showed no significant differences in IL-8 production, while *S. typhimurium* showed a fourfold increase compared to cells with no bacterial exposure (Fig. 3d). Overall, the transwell assays indicated that neither PBP8, nor the expression of wt-CsgA or CsgA-TFF3 led to any phenotypes that compromised epithelial integrity compared to unmodified EcN.

**Transient colonization and curli production of modified EcN**. Our initial in vivo experiments focused on demonstrating the viability and persistence of engineered EcN strains in the mouse GI tract after oral administration. For these experiments, we employed

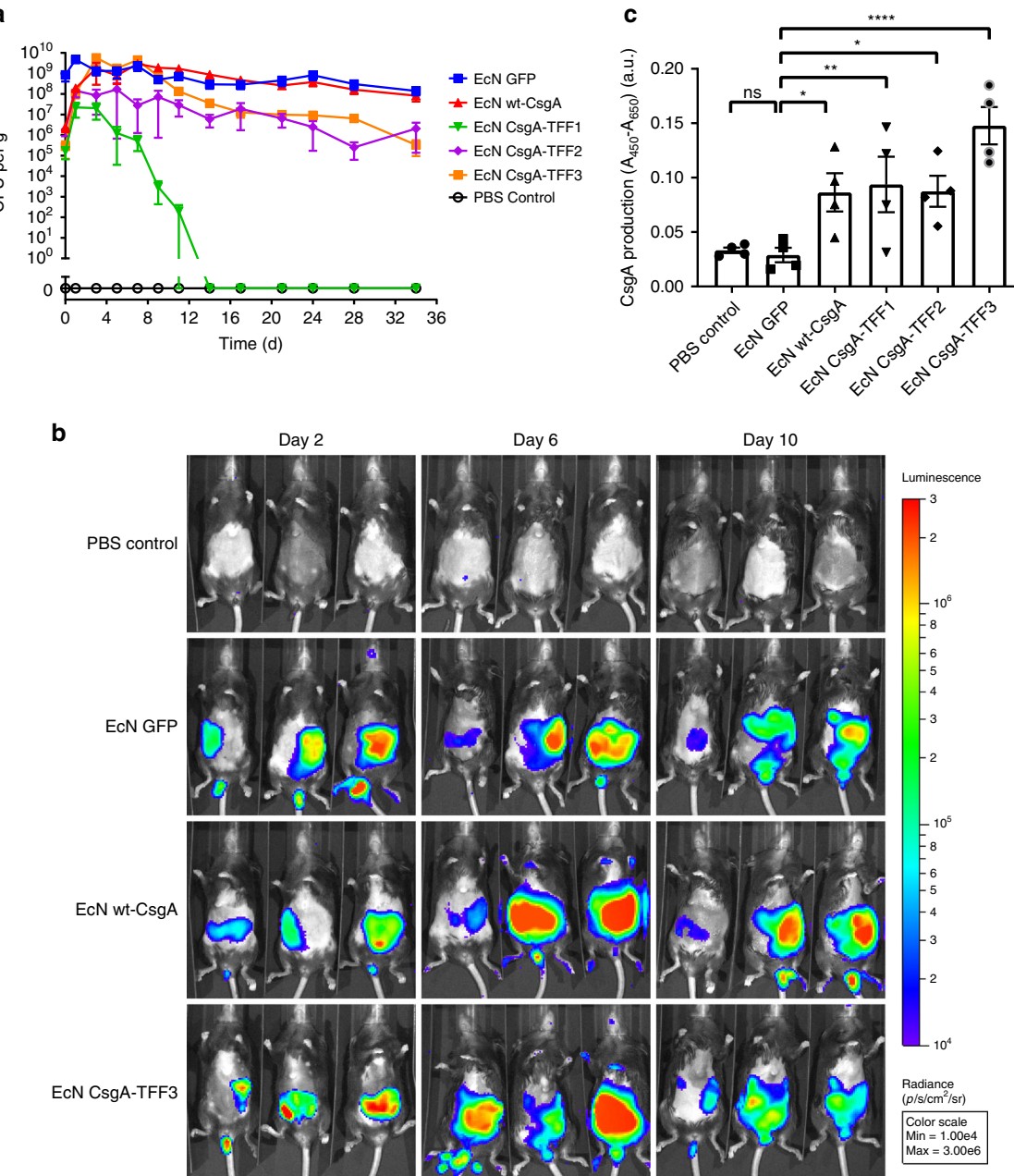

**Fig. 4 Residence time of engineered EcN in the mouse gut and in vivo curli expression. a** In vivo residence time for EcN variants, as measured by CFU counting from fecal samples (Blue = EcN GFP, Red = EcN wt-CsgA, Green = EcN CsgA-TFF1, Pink = EcN CsgA-TFF2, Orange = EcN CsgA-TFF3, Black = PBS Control). **b** IVIS images of mice that received either PBS or luminescent EcN variants at various time points post-inoculation. **c** Relative CsgA production from fecal sample analysis of mice 5 days post-inoculation, as measured by ELISA. Data are represented as mean ± SEM. (ns) $p > 0.05$, *$p \le$ 0.05, **$p \le 0.01$, ***$p \le 0.001$, ****$p \le 0.0001$, one-way ANOVA followed by Fischer's Least Significant Difference (LSD) multiple comparison. Source data are provided as a Source Data file.

an EcN strain with a genomically integrated luminescence operon (*luxABCDE*) to facilitate in vivo tracking[37,38]. After transformation of this EcN strain with the panel of plasmids encoding the synthetic curli operons, the strains were administered to healthy mice (C57BL/6NCrl) via oral gavage, concurrent with drinking water containing kanamycin and L-(+)-arabinose in order to maintain the plasmids and induce curli expression. A single dose of $10^8$ colony-forming units (CFU) led to persistent colonization (>32 days) of the mouse GI tracts for all but one of the curli producing strains (EcN wt-CsgA, EcN CsgA-TFF2, and EcN CsgA-TFF3), as measured by CFU counted from fecal samples. Notably, EcN wt-CsgA maintained a CFU count of $10^8$–$10^9$ CFU g$^{-1}$ over

the course of the experiment, similar to that of EcN transformed with a pBbB8k plasmid encoding for green fluorescent protein (GFP) as a control (Fig. 4a). This suggested that wild-type curli fiber overproduction in vivo did not compromise the fitness of the engineered EcN any more than recombinant production of any heterologous intracellular protein. In comparison, EcN CsgA-TFF2 and EcN CsgA-TFF3 concentrations fell over the course of the experiment to ~$10^6$–$10^7$ CFU, suggesting that production of the CsgA-TFF fusions was stressful enough to compromise the colonization ability of EcN in the stringent environment of the mouse gut. We speculate that this may be due to the large size of the TFF fusion domains[39], in addition to their propensity to form internal

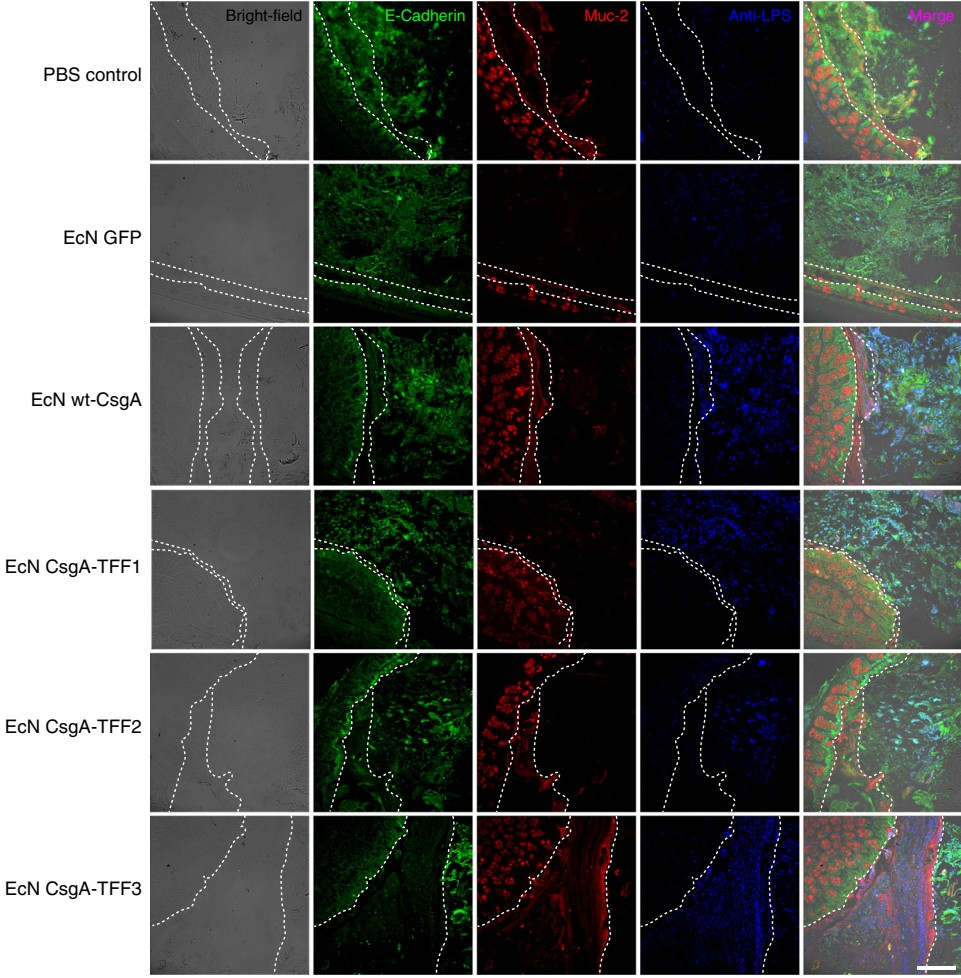

**Fig. 5 Immunohistological visualization of engineered EcN strains in tissue sections.** These sections are taken from proximal colons of mice receiving different bacteria. Sectioning protocol was designed to preserve mucus and luminal content. Sections were stained with fluorescently labeled antibodies: anti-E-cadherin (green), anti-Muc2 (red), and anti-LPS (blue). The first column shows bright-field images of the sections. The last column shows an overlay of all stains. The white dotted lines represent the boundary of the epithelium and mucus layers. The leftmost parts represent the epithelium, the center parts represent the mucus layers and the rightmost parts represent the lumen (scale bar = 100 μm).

disulfide bonds that could hinder extracellular export. Indeed, this difference between wt-CsgA and CsgA-TFF-producing strains is reflected in the corresponding in vitro growth rates (Supplementary Fig. 3). We further confirmed the presence of the engineered EcN strains in living animals by visualizing bacterial luminescence on days 2, 6, and 10 after oral administration using an in vivo imaging system (IVIS). As expected, the luminescence of the EcN could be observed in the abdomen of all mice that had received bacteria (Fig. 4b and Supplementary Fig. 4).

In addition to the engineered bacteria themselves, we sought to confirm that the engineered curli fibers were being produced in situ. We therefore performed ELISA using an anti-6xHis antibody on homogenized fecal samples obtained 5 days after oral administration. We found that, as expected, only mice that had received curli producing strains (EcN wt-CsgA and EcN CsgA-TFFs) showed signal above background levels (Fig. 4c). In comparison, EcN producing GFP showed only background signal. Finally, we used immunohistochemistry at the experimental endpoint to visualize the engineered bacteria directly in tissue sections of the proximal and distal colon (Fig. 5). The bacteria were probed with a fluorescently labeled anti-lipopolysaccharide (LPS) antibody specific to *E. coli* (blue), while anti-MUC2 (red), and anti-E-cadherin (green) stains were used to visualize the mucins and colonic epithelial cells, respectively. The bright-field images help identify the

luminal boundary. We found that due to the fixing and staining protocol, which were designed to preserve the integrity of the intestinal mucins, the E-cadherin antibodies stained not only the colonic cells, but also cytoskeletal elements found in the feces and other contents of the gut lumen present in the tissue sections. Nevertheless, the distinct pattern and arrangement of colonic cells in the sections could be used to identify the epithelial border and differentiate them from luminal content. The anti-LPS staining revealed that EcN, EcN wt-CsgA, and EcN CsgA-TFFs could be observed throughout the gut lumen and near the most superficial layers of mucus, while the PBS control showed minimal background staining. Interestingly, the CsgA-TFF3 signal co-localized with the MUC2 signal, suggesting that the mucin-binding activity of the TFF3 promoted mucus integration of the engineered curli fibers.

**Amelioration of disease activity in a mouse model of colitis.** Based on the ability of the engineered EcN strains to colonize the mouse GI tract and the previously demonstrated wound healing and mucin-binding activity of CsgA-TFF3 curli fibers in vitro[24], we sought to investigate their efficacy in a dextran sodium sulfate (DSS) model of murine colitis[40,41]. DSS is a chemical colitogen that can be administered orally to induce epithelial damage and compromised barrier function in the mouse colon. Unmodified

EcN and TFF secretion from *Lactococcus* spp. have each individually shown some efficacy against the DSS model, so we reasoned that their combination with the PATCH system would have similarly beneficial effects[15].

We examined the protective effects of engineered CsgA-TFF3 produced by PBP8 using female C57BL/6NCrl mice that had been administered 3% DSS over 5 days to induce colonic inflammation. Pilot experiments with oral administration of PBP8 strains were not very effective in decreasing disease symptoms. However, histological analysis and further literature consultation revealed that DSS-induced colitis was most severe in the distal colon, whereas the engineered bacteria resided mostly in the cecum and proximal colon (Supplementary Fig. 5)[29,42,43]. In order to circumvent this peculiarity of the murine DSS model and investigate the efficacy of our approach, we pivoted to rectal administration of the bacteria so that they could easily co-localize with the affected tissues. Notably, we do not envision that this issue would affect the efficacy of engineered bacteria in other models or in humans, as both oral and rectal deliveries are viable routes of drug administration depending on the patient's disease localization.

As outlined in Fig. 6a, the mice received daily administrations of PBP8 rectally for 3 days prior to DSS intake, during 5 days of DSS intake, and during a 5-day recovery period. DSS treatment in mice that had not received any bacteria (PBS DSS+) led to intestinal inflammation that could be observed by weight loss and increases in disease activity index (DAI, a composite measure of weight loss, diarrhea index and rectal bleeding, Table 1) compared to mice in the healthy control group, without DSS treatment (PBS DSS−). Mice that received bacteria expressing CsgA-TFF3 (PBP8 CsgA-TFF3 DSS+) had significantly ameliorated weight loss and reduced DAI over the course of the experiment and almost returned to the same state as the healthy control group 5 days after DSS removal (Fig. 6b, c). Mice that received engineered bacteria that were either producing cytosolic GFP as a control (PBP8 DSS+) or producing wild-type CsgA (PBP8 wt-CsgA DSS+) showed results similar to the disease group without any bacterial administration. The mice were sacrificed at day 10 for investigation of their colons to examine the effects of PBP8 CsgA-TFF3 on the induced colitis. DSS inflammation is associated with colon length reduction[44]. We found that colon length did not differ significantly between the PBP8 CsgA-TFF3 group and the healthy control group, suggesting that in situ CsgA-TFF3 production attenuated colonic inflammation caused by DSS. In comparison, the colitic control, PBP8, and PBP8 wt-CsgA groups all had shorter colons (Fig. 6d).

We also assessed the effects of bacterial administration on gut inflammation using histology. Common histological characteristics of DSS-induced colitis include immune cell infiltration involving multiple tissue layers, as well as loss of colonic crypts and epithelial damage[45,46]. We employed a histological scoring system based on previously published accounts (Table 2)[45–47]. All of the groups that received DSS treatment had significantly higher histopathology scores than the healthy control group, indicating more inflammatory effects (Fig. 6e). The histology score for the PBP8 CsgA-TFF3 group was the lowest of all the groups that received DSS, though the difference was not quite significant according to our scoring and statistics criteria. Images of the histology sections from the colitic control group showed complete loss of crypt structure, goblet cell depletion, immune cell infiltration into the lumen, and edema of the tissues around the colon, all of which reflect the severe inflammatory effects of DSS treatment (Fig. 6f–j). In contrast, tissue sections from mice treated with bacteria showed some qualitative improvements, with better preservation of crypt structure. The PBP8 CsgA-TFF3 group also showed lower inflammatory cell infiltration, less

edema, and more intact epithelium. One explanation for the histological similarity among the DSS treated groups could be that the tissue sections were obtained 5 days after DSS treatment had stopped. Therefore, natural healing processes could have obscured any quantitative differences in histopathology between the groups that received DSS and bacteria.

In order to assess the effects of CsgA fusion on the bioactivity of TFF3, we performed a trial experiment with a strain of PBP8 engineered to secrete TFF3 in a soluble form. PBP8 was transformed with pBbB8k-N22-TFF3, a plasmid encoding the N22 secretion signal peptide followed by the TFF3 encoding sequence. Thus, the PBP8 N22-TFF3 strain secreted TFF3 in a soluble form through the same curli secretion machinery as CsgA-TFF3. Using an identical protocol to the original DSS-induced colitis experiment (Fig. 6), we ran a smaller experiment with mice randomly assigned to one of three experimental groups: colitic group (PBS DSS+), DSS treated soluble TFF3 group (PBP8 N22-TFF3 DSS+) and DSS treated curli-bound TFF3 group (PBP8 CsgA-TFF3 DSS+). The results of this experiment mirrored those of the original—CsgA-TFF3 continued to receive the lowest histology scores, though the difference between the groups was not statistically significant based on our metrics (Supplementary Fig. 6). This is likely due to high variability in the DSS model, since the mice in the colitic group did not become injured as severely as they did in the original experiment, despite the identical protocol.

**Immunomodulation and barrier enhancement from modified EcN**. In order to start probing the mechanism of the apparent protective effects of the PATCH system with CsgA-TFF3, we analyzed gene expression and protein production profiles from colonic tissue homogenates across the experimental groups. We used a parallelized ELISA assay (Luminex), to probe cytokine levels from tissue homogenates obtained from mice at the experimental endpoint. We found that IL-6, IL-17A, and TNF-α concentrations decreased in colonic tissues for the PBP8 CsgA-TFF3 group compared to the colitic group (Fig. 6k–l and Supplementary Fig. 7A). It is worth noting that IL-17A shows a profound difference when comparing PBP8 CsgA-TFF3 even to the other groups, including PBP8 wt-CsgA. IL-1β concentrations were also lower, but not statistically significant (Supplementary Fig. 7B). Cytokines such as TGF-β, IL-10 and IFN-γ were not significantly affected by any of the bacterial treatments (Supplementary Figs. 7C, D and 8G). The affected cytokines (IL-6, IL-17A, TNF-α, and IL-1β) directly relate to differentiation and secretion of T helper 17 (Th17) cells, which is a lineage of CD4+ cells that mediates innate and adaptive immunity against various pathogens at mucosal sites[48].

The DSS colitis model leads to several other changes, including downregulation of genes associated with epithelial barrier function and upregulation of genes associated with inflammatory signaling[49,50]. With respect to tight junction protein-1 (TJP-1), a.k.a. zona occludens-1 (ZO-1), the PBP8 CsgA-TFF3 group showed significantly higher messenger RNA (mRNA) levels than the colitic control group (Supplementary Fig. 8A). This is in line with the known functions of TFFs in mammalian hosts even though other markers of barrier function associated with TFF bioactivity (occludin, claudin-2, and intestinal TFF3, Supplementary Fig. 8D–F) did not show changes according to quantitative real-time reverse transcription PCR (qRT-PCR) analysis. Notably, the PBP8 wt-CsgA group also exhibited high *tjp-1* mRNA levels. This could the explained by known interactions between wt-CsgA and toll-like receptor 2 (TLR2), which can indirectly lead to moderate increases in *tjp-1* expression[51,52]. Indeed, in vitro experiments on polarized Caco-2 monolayers aimed at better

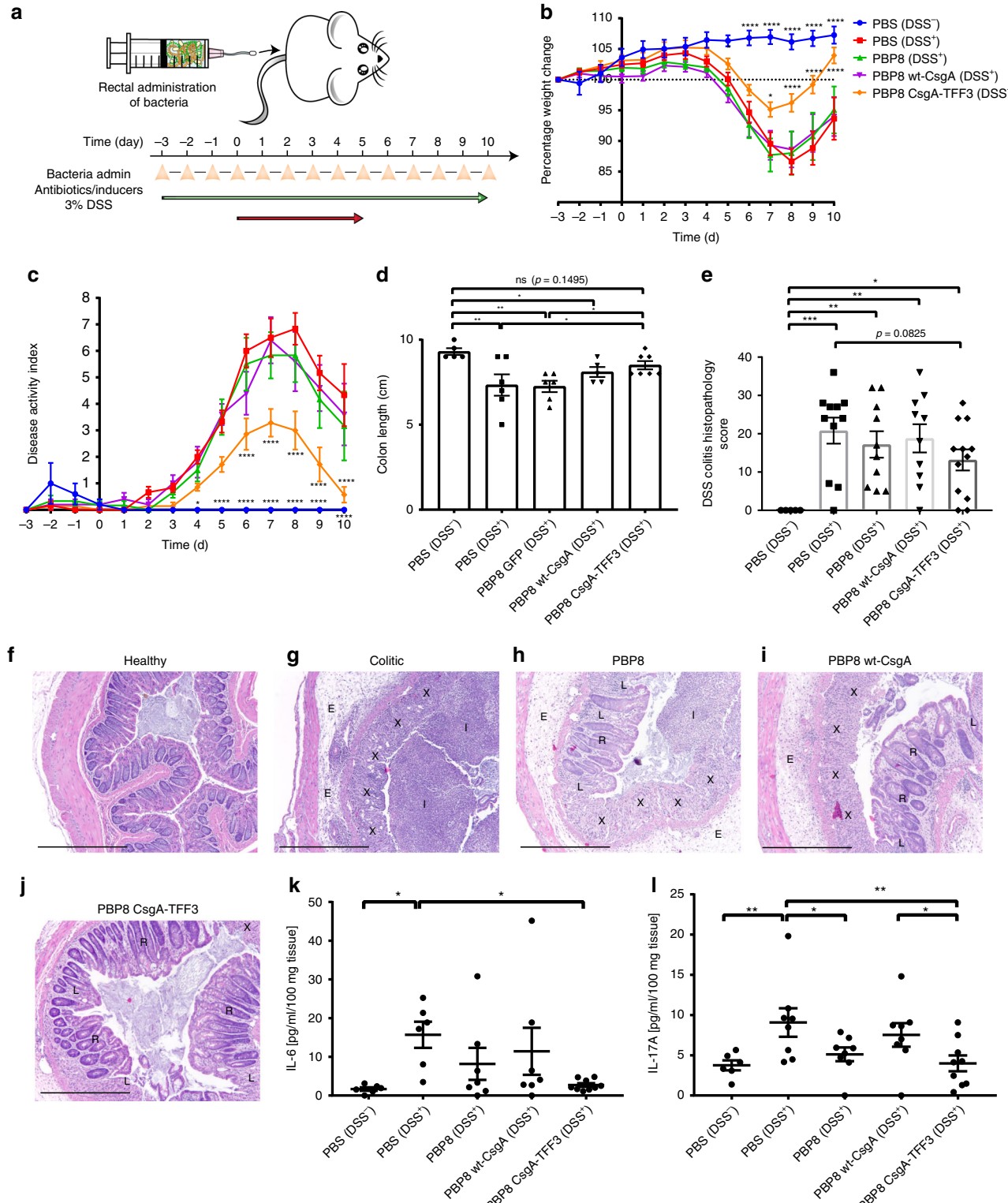

characterizing the barrier-enhancing function of CsgA-TFF3 led to similarly inconclusive results when compared to wt-CsgA (Supplementary Fig. 9). Although the effects of TFF3 fusion to CsgA on its interaction with host receptors is not clear from this work, it is possible that both domains could contribute to modulating local gene expression.

Matrix metalloproteinase-9 (MMP-9), whose function is to breakdown extracellular matrix proteins in inflammatory environments, is upregulated during IBD and is an essential mediator

of tissue injury during DSS colitis. Blockade of MMP-9 has also been explored as a clinical treatment for IBD[53]. In our experiments, the colitic control and PBP8 control groups showed elevated *mmp9* mRNA levels compared to the healthy control group, while the PBP8 CsgA-TFF3 group showed significantly lower *mmp9* expression (Supplementary Fig. 8B). Prostaglandin-endoperoxidase synthase 2 (PTGS-2), a.k.a. cyclooxygenase-2 (COX-2), showed similar results to MMP-9, with the PBP8 CsgA-TFF3 group significantly lower than the colitic group, and

**Fig. 6 Therapeutic efficacy of engineered EcN against a mouse model of DSS-induced colitis. a** Schematic of administration schedule. PBP8 variants ($10^8$ CFU) were administered rectally daily. Antibiotics and inducers were administered continuously via drinking water. Weight change (**b**, $N = 9$) and disease activity index (**c**, $N = 5$–7) were monitored over time, averaged across two independent experiments. Activity index criteria are described in Table 1 (Blue = PBS (DSS−), Red = PBS (DSS+), Green = PBP8 (DSS+), Pink = PBP8 wt-CsgA (DSS+), Orange = PBP8 CsgA-TFF3 (DSS+)). **d** Colon length at endpoint from two independent experiments ($N = 5$–7). **e** Combined DSS colitis histopathology score reflecting severity of inflammation (see Table 2 for details), at endpoint from two independent experiments ($N = 5$–10). **f–j** Representative histology of distal colon sections stained with haematoxylin and eosin from each experimental group: **f** PBS (DSS−) – healthy control, **g** PBS (DSS+) – colitic, **h** PBP8 (DSS+), **i** PBP8 wt-CsgA (DSS+), **j** PBP8 CsgA-TFF3 (DSS+). Image markers indicate complete loss of crypt and goblet cell depletion (X), immune cell infiltration (I), tissue edema (E), partial loss of the crypts (L), and recovery of crypts (R) (scale bar = 500 μm) **k–l** IL-6 and IL-17A protein levels from tissue homogenates, as determined by multiplex ELISA. Data are presented as protein concentration per 100 mg of tissue from two independent experiments ($N = 6$–9). Data are represented as mean ± SEM. (ns) $p >$ 0.05, $*p \leq 0.05$, $**p \leq 0.01$, $***p \leq 0.001$, $****p \leq 0.0001$. The time course experiments (**b** and **c**) were analyzed using two-way ANOVA following by Dunnett's multiple comparison. One-way ANOVA followed by Fischer's LSD multiple comparison was used for **d**, **e**, **k**, and **l**. Source data are provided as a Source Data file.

**Table 1 Disease activity index (DAI) parameters and their associated scoring schemes.**

| Score | Weight loss (%) | Stool consistency | Blood in stool |
|---|---|---|---|
| 0 | None | Normal | Normal |
| 1 | 1–5 | Slightly loose stool | Small presence of blood |
| 2 | 5–10 | Loose stool | Significant presence of blood |
| 3 | 10–15 | Diarrhea | Gross blood |
| 4 | >15 | | |

**Table 2 Histological grading scheme for DSS colitis.**

| Parameter graded | Score | Description |
|---|---|---|
| Severity of inflammation | 0 | None |
| | 1 | Slight |
| | 2 | Moderate |
| | 3 | Severe |
| Depth of injury | 0 | None |
| | 1 | Mucosa |
| | 2 | Mucosa and submucosa |
| | 3 | Transmural |
| Crypt damage | 0 | None |
| | 1 | Basal one-third damaged |
| | 2 | Basal two-third damaged |
| | 3 | Only surface epithelium intact |
| | 4 | Entire crypt and epithelium lost |
| Percent involvement | x1 | 0–25% |
| | x2 | 26–50% |
| | x3 | 51–75% |
| | x4 | 76–100% |

comparable with the healthy control group (Supplementary Fig. 8C). PTGS-2 is an enzyme whose production is induced in the colon during active IBD[54]. Its inhibition has also been investigated as an anti-inflammatory treatment, and has ameliorated colitis in a murine DSS model[55]. Similar to the case of MMP-9, the PBP8 wt-CsgA group also showed reduced *ptgs2* expression, which likely reflects the known interactions of unmodified curli fibers with the gut epithelium observed in this work and elsewhere[56].

## Discussion

We have developed an engineered beneficial bacterium that produces a self-assembled matrix in situ within the GI tract with programmable functions. Our engineering strategy (PATCH) enabled the production of a modified curli fiber matrix with fused TFFs under physiological conditions in a manner that does not appear to alter EcN's inherent lack of pathogenicity. Administration of the engineered bacterium before, during, and after the induction of colitis in mice led to amelioration of inflammation and a reduction in Th17 responses in the colon. The protective effect of PATCH with CsgA-TFF3 also correlated with enhanced mucosal barrier function and is correlated with a reduction in the expression of inflammatory cytokines and enzymes in colonic tissues. More detailed studies will be required to rigorously evaluate the relative mechanistic contributions of the native CsgA and the fused TFF3 domains using this system. Nevertheless, results from the genetic and biochemical analyses we performed were in line with previous accounts of trefoil factor bioactivity, which includes immunomodulation, the promotion of epithelial restitution, and upregulation of tight junction proteins (Fig. 1)[7,23].

Ongoing work in the lab is focused on improving the PATCH system in order to make it more suitable for clinical deployment. In the experiments described here, antibiotics and small molecule inducers were fed to the mice in order to maintain the plasmid's stability in the modified EcN strains and induce curli production, respectively. The antibiotics alter the gut microbiome significantly[57]. To address this issue, second generation PATCH systems should be engineered with stable plasmid systems that do not require antibiotics that have been reported on extensively elsewhere[38,58,59]. Otherwise, the PATCH system could potentially be integrated into the genome of the bacteria, though the low copy number of the genes might result in lower amounts of therapeutic production. Regarding the inducers, we can replace the arabinose inducible promoter with environmentally sensitive promoters that respond to temperature or inflammatory markers to avoid the use of external inducers and further improve the system.

Although we demonstrated that oral administration was a viable approach to establishing a detectable and stable population of engineered EcN strains in the murine lower intestine, we ultimately opted for rectal administration to co-localize the bacteria near diseased areas and demonstrate efficacy. However, human IBD pathology can be spread across the colon and parts of the small intestine. Furthermore, *E. coli* is similarly localized in humans, and is known to proliferate at sites of inflammation[60]. Therefore, either oral or rectal delivery may be relevant for other models of IBD or in humans.

We also recognize that the curli fibers themselves are not a blank slate material in that CsgA already has numerous known interactions with host cells and tissues that could confound the effects of the displayed domains. In fact, wild-type curli fibers were shown to have some anti-inflammatory and barrier protective properties, which could confound our ability to observe the effects of the appended trefoil factors[31,52,56,61]. While this

may or may not impede further development of PATCH with curli fibers as a scaffold, we know that the biosynthetic machinery dedicated to curli secretion can tolerate a wide range of heterologous proteins[62]. We are therefore in the process of exploring other combinations of scaffolding proteins and bioactive domains that can be secreted through the curli (a.k.a. Type VIII) pathway to circumvent these confounding effects and probe different therapeutic modalities.

We chose the acute DSS-induced colitis model because of its practical accessibility and its appropriateness for studying mucosal healing in the mammalian GI tract. However, the potential applicability of PATCH for the treatment of diseases like IBD will require further studies in complementary disease model systems (e.g., IL-10 knockout, adoptive T-cell transfer, TNBS injury)[40]. It is worth noting that when comparing our results with current literature on the use of traditional, oral anti-inflammatory drugs such as 5-aminosalicylic acid (5-ASA) for DSS colitis murine model treatment[63–65], we found that our PATCH technology seemed to improve the weight, DAI, and colon length of the mice to a better extent than 5-ASA. Future studies include side-by-side comparison and combination of PATCH and 5-ASA would be interesting to pursue as a multi-pronged approach toward alleviating colitis. Ongoing work in our lab is focused on probing the difference between the secretion of various curli-tethered (via fusion to CsgA) and soluble therapeutic proteins. Although we did not observe a difference between tethered and untethered TFF3 in this work, the multivalency and mucoadhesion offered by the CsgA-TFF3 scaffold could be advantageous for increasing the local concentration of drugs in the gut[66]. Indeed, future iterations of the PATCH platform may allow for synergy between tethered and soluble therapeutic domains.

## Methods

**Cell strains and plasmids.** *E. coli* Nissle 1917 Prop-Luc strain (EcN *LuxABCDE*, erythromycin resistance) was kindly provided as a gift from Sangeeta Bhatia's lab (Massachusetts Institute of Technology). *E. coli* PBP8 strain was derived from *E. coli* Nissle 1917 by genomic deletion of the curli operon[29]. *S. typhimurium* (strain SL1344) was provided by Pam Silver's lab (Harvard University).

The design and construction of the synthetic curli operon encoding plasmids are described in detail elsewhere[29]. Briefly, a pBbB8k plasmid backbone contains the genes csgBA*CEFG as a single cistron, controlled by the araBAD promoter, where A* indicates either wild-type or chimeric CsgA. Gene fragments encoding 6xHis tag modified TFF1-3 domains were cloned into these vectors to create pBbB8k-CsgA-TFF1, pBbB8k-CsgA-TFF2, pBbB8k-CsgA-TFF3, and pBbB8k N22-TFF3 (Supplementary Tables 1 and 2 for CsgA fusion sequences, Supplementary Table 3 for cloning and sequencing primer sequences, and Supplementary Fig. 10 for pBbB8k-wt-CsgA plasmid map). The list of bacteria strains and plasmids can be found in Supplementary Table 4. The list of reagents with distributers can be found in supplementary reagent list document.

**Mice.** Female 8- to 9-week-old C57BL/6NCrl mice were obtained from Charles River Laboratories. Mice were accommodated in SPF conditions while sterile food (normal mouse chow, LabDiets 5K67, or non-fluorescent food, LabDiets 5V5R) and water were provided ad libitum. Sterile vinyl isolators equipped with food, water and bedding were used to house the mice within the Harvard Medical School animal facility. Mice had at least 1 week of acclimatization to the facility environment before any experiment. All experiments were conducted in compliance with US National Institutes of Health guidelines and approved by the Harvard Medical Area Standing Committee on Animals.

**Cell culture.** Caco-2 cells (C2BBe1 [clone of Caco-2] (ATCC® CRL2102™)) were maintained and passaged in Dulbecco's modified Eagle medium (DMEM) supplied with 4.5 g L$^{-1}$ glucose and glutamax, 15% fetal bovine serum (FBS) and 1% Penicillin–Streptomycin (Gibco) in a humidified incubator at 37 °C, 5% CO$_2$. For the translocation assay, epithelial integrity and IL-8 production experiments, Caco-2 cells were grown to confluence on 3.0 µm semipermeable tissue culture inserts (24 wells, Transwell, Corning). After 14–21 days, the cell monolayers achieved a polarized, differentiated state and the transepithelial electrical resistance (TEER) reached 1000–1200 Ω cm$^{-2}$.

**In vitro expression of engineered curli fiber.** EcN Prop-Luc or PBP8 cells were transformed with the corresponding pBbB8k plasmids by electroporation to create variants of curli producing cells and plated onto LB agar plates containing 50 µg mL$^{-1}$ kanamycin (Teknova) and incubated overnight at 37 °C. Individual colonies were inoculated in 5 mL of LB media containing 50 µg mL$^{-1}$ kanamycin, and grown overnight in 37 °C shaking incubator. Overnight starter cultures were diluted 100-fold in new LB media containing 50 µg mL$^{-1}$ kanamycin at desired volumes and incubated while shaking at 37 °C until the refreshed cultures reached a log phase at an optical density (OD) at 600 nm of 0.5 to 0.8. Then, protein expression was induced by adding L-(+)-arabinose to a final concentration of 0.05% (weight per volume). The induced cultures were grown in 37 °C shaking incubator overnight to allow protein expression.

**Quantitative Congo Red-binding assay.** One milliliter of bacterial culture was pelleted at 8000 rpm for 10 min and resuspended in a 0.025 mM solution of Congo Red (CR) in phosphate-buffered saline (PBS) for 10 min. After pelleting the cells again at 14,000 rpm for 10 min, the absorbance of the supernatant at 490 nm was measured using a microplate reader. Normalized curli fiber production was calculated by subtracting the measured absorbance value from that measured for 0.025 mM Congo Red in PBS and normalized by the OD$_{600}$ of the original bacterial culture.

**Whole-cell filtration ELISA.** The bacterial cultures were diluted to OD$_{600}$ of 0.3 with tris-buffered saline (TBS). The specimens (200 µL) were transferred to a Multiscreen-GV 96-well filter plate, filtered, and washed with TBST buffer (TBS, 0.1% Tween-20). After blocking with with 1% bovine serum albumin (BSA) and 0.01% H$_2$O$_2$ in TBST for 1.5 h at 37 °C, and subsequent washing steps, 50 µL of anti-6xHis antibody-horseradish peroxidase (HRP) conjugate (1:200, Thermo Fisher, MA1-80218) was added to each well and incubated for 2 h at 25 °C. For the TFF3 antibody-binding assay, samples were incubated with 50 µL of anti-TFF3 primary antibody (1:450, Sigma, WH0007033M1) for 2 h at 25 °C and washed three times with TBST buffer, followed by incubation of 100 µL goat anti-mouse-HRP conjugated secondary antibody (1:5000, Thermo Fisher, 31430) for 1 h at 25 °C and three subsequent washes. After the wash steps, Ultra-TMB (3,3′,5,5′-tetramethylbenzidine) ELISA substrate (100 µL) was added to each well and incubated for 10 min at 25 °C. To stop the reaction, 50 µL of 2 M sulfuric acid was added to each well. One-hundred microliters of the final reaction was transferred to 96-well plate and measured the absorbance at 450 and 650 nm. The relative amount of displayed peptide was measured by subtracting absorbance at 450 nm with absorbance 650 nm.

**Electron microscopy.** Two-hundred microliters of the testing cultures were filtered onto Nucleopore Track-Etched membranes (0.22 µm pore size) under vacuum and placed in fixative solution (2% glutaraldehyde and 2% paraformaldehyde in 0.1 M sodium cacodylate buffer) for 2 h at room temperature. After fixation, the membranes were gently rinsed with water and subjected to an ethanol gradient (25, 50, 75, 100, and 100% (volume per volume)) with a 15 min incubation for each concentration. The samples were then transferred to a critical point dryer). The dried membranes were placed on Scanning Electron Microscopy sample holders with carbon adhesives and sputter coated with 80:20 Pt:Pd (5 nm-thick). A Zeiss Ultra 55 Field Emission Scanning Electron Microscope was used to image the samples.

**Invasion assay.** Caco-2 cells at passage 5–15 were plated in 24-well plates at a density of 10$^5$ cells per well in 500 µL of regular cell culture media and grown to 90% confluency. The bacterial cultures were pelleted, washed with PBS and diluted to an OD$_{600}$ of 0.5 in DMEM with 1 g L$^{-1}$ glucose and 1% FBS. The Caco-2 cells were rinsed twice with PBS to remove the antibiotic before addition of the bacteria (500 µL). Bacteria were incubated for 2 h with the Caco-2 cells and removed by aspiration. The Caco-2 cells were then washed twice with 500 µL PBS before receiving 500 µL of DMEM, 1 g L$^{-1}$ glucose, 1% FBS, and 100 µg mL$^{-1}$ gentamicin. After 1 h of incubation, the media were aspirated and replaced with 1 mL of 1% Triton-X. The cells were incubated with Triton-X at 37 °C for 10 min and repeatedly pipetted to homogenize. Each well was serially diluted and plated on kanamycin plates to count the colony-forming units (CFU) of bacteria that had invaded the Caco-2 cells.

**Translocation assay.** EcN variants and *S. Typhimurium* SL 1344 (Silver lab, Harvard Medical School) were grown and induced, if applicable, in LB media with appropriate antibiotics, pelleted, washed with PB,S and diluted to OD$_{600}$ of 0.01 in DMEM with 1% glucose and 1% FBS. One day before the experiment, the culture media of polarized Caco-2 (See additional materials and methods) were switched to their corresponding non-antibiotic counterparts with 1% glucose and 1% FBS. Six-hundred microliters of diluted cultures were used to infect polarized Caco-2 apically. After 5 h of incubation, 100 µL of media were collected from apical and basolateral sides of the transwell, serially diluted and plated on antibiotic selective plates to enumerate percentage of translocated bacteria.

**Epithelial integrity**. The epithelial integrity of polarized Caco-2 was determined by TEER values, and a fluorescein isothiocyanate-labeled dextran (FITC-dextran) translocation experiment. Prior to the infection, the TEER value of each polarized Caco-2 well was determined using Millicell ERS-2 Voltohmmeter. The cells were infected in the same manner as the translocation assay protocol. At 24 h post infection, the TEER value was measured again to calculate the reduction of TEER. At the same time, 5 μL of 10 mg mL$^{-1}$ FITC-dextran (average molecular weight 3–5 kDa) was added apically to each transwell. Two hours after the addition, 100 μL of media from the apical and basolateral sides were collected and transferred to a black, clear bottom, 96-well plate and the fluorescence intensity was measured using a plate reader at 485 nm excitation and 520 nm emission wavelength.

**IL-8 production**. Polarized Caco-2 cells were infected in the same manner as the translocation assay. At 24 h post infection, 100 μL of media from the basolateral side was collected and assayed to determine the IL-8 concentration using ELISA kit.

**In vivo residence time study**. This protocol was approved by the Harvard Medical Area Standing Committee on Animals (HMA IACUC) (Ref. No. 05185). Female 8- to 9-week-old C57BL/6NCrl mice were randomly assigned into six groups that each received a different variant of EcN Prop-luc bacteria: GFP, wt-CsgA, CsgA-TFF1, CsgA-TFF2, CsgA-TFF3, and PBS control. Mice were subjected to 18 h of fasting prior to the experiment, to remove the food in the upper GI tract, but received L-(+)-arabinose (10 g L$^{-1}$) and kanamycin (1 g L$^{-1}$) via drinking water. At the beginning of the experiment (day 0), the bacterial samples were prepared by culturing them to log phase and concentrating them to OD$_{600}$ of 10 in 20% sucrose in PBS. Mice were given either 100 μL of PBS or bacterial suspensions based on their specified groups through oral gavage. Afterward, mice were returned to normal chow with L-(+)-arabinose and kanamycin drinking water. At day 0 (5 h after the administration), 1, 3, 5, 7, 9, 11, 14, 17, 21, 24, 28, and 34, fecal samples from each mouse were collected, weighed, serially diluted, and plated on antibiotic selective plates (erythromycin and kanamycin) to enumerate resident CFU over time.

**In vivo imaging of engineered microbes**. Mice were given an alfalfa-free chow for at least 5 days prior to the experiment to minimize autofluorescence. Mice were given the engineered EcN in the same manner as the residence time protocol in terms of special water and fasting regime. At day 2, 6, 10, mice were shaved at the abdominal area to help expose the GI tract to the IVIS machine. Mice were then imaged under anesthesia using IVIS Lumina II using luminescence filter, with field of view (FOV) = D (12.5 cm), fstop = 1 and large binning. Living Image software version 4.3.1/4.4 was used for image analysis. The detailed protocol for curli immunohistochemistry can be found in additional materials and methods.

**Fecal filtration ELISA**. Homogenized fecal samples from day 5 of the residence time experiment were transferred onto the Multiscreen-GV 96-well filter plate (0.22 μm pore size). The volumes were normalized so that about 1.25 mg of fecal samples were on each membrane. Then, the filter plate underwent similar processes: blocking, incubating with Anti-6xHis antibody-horseradish peroxidase (HRP) (1:200, Thermo Fisher, MA1-80218), washing and interacting with the TMB substrate, as the whole-cell ELISA protocol described above.

**Curli immunohistochemistry**. Six groups of mice: PBS control, EcN GFP, EcN wt-CsgA, EcN CsgA-TFF1, EcN CsgA-TFF2, and EcN CsgA-TFF3 were treated in a similar manner in the residence time study. Three days after the oral administration of bacteria, mice were sacrificed to collect the colonic tissues, which were divided into three 1-cm sections: proximal, medial, and distal colons. The tissues were cassetted and fixed in Carnoy's solution (100% ethanol, chloroform and glacial acetic acid in a 6:3:1 ratio), to preserve the mucus layer, for 3 h at room temperature and transferred to PBS. The fixed tissues were embedded in paraffin and sectioned onto the slides prior to immunohistochemistry steps.

The triple immunofluorescence antigen labeling of the paraffin-embedded murine colon samples was performed with anti-LPS, Mucin-2 and E-cadherin antibodies. The paraffin sections were deparaffinized and rehydrated, followed by antigen retrieval using sodium citrate buffer (pH 6). The sections were then incubated with 1 mg per milliliter sodium borohydride (ICN chemicals) for 5 min at room temperature. After three washes with TBS, the sections were incubated with Mouse On Mouse (M.O.M.) blocking kit (Vector Labs BMK-2202) for an hour at room temperature. After three washes with TBS, the sections were incubated with 5% normal donkey serum (Jackson ImmunoResearch Lab Inc, West Grove PA) for an hour at room temperature. Slides were then incubated with mouse anti-LPS (1:200, Abcam ab35654), rabbit anti-Mucin-2 (Santa Cruz sc-15334) and goat anti-E-cadherin (1:200, R&D Systems AF648) overnight at 4 °C. The slides were washed three times and incubated with Alexa Fluor 647 conjugated Donkey anti-mouse secondary antibody (1:300, Invitrogen), Cy3 conjugated Donkey anti-rabbit secondary antibody (1:300, Jackson ImmunoResearch Lab), and Alexa Fluor 488 conjugated Donkey anti-goat secondary antibody (1:300, Invitrogen). Samples were counterstained with 4′,6-diamidino-2-phenylindole

(DAPI) and then washed three times with TBS, and the slides were mounted with Prolong Gold anti-fade mounting media (Invitrogen).

**DSS model of mouse colitis and treatment protocol**. Mice were randomly assigned to five groups: non-colitic (PBS DSS$^-$), colitic (PBS DSS$^+$), PBP8 with GFP control vector-treated (PBP8 DSS$^+$), PBP8 with wild-type CsgA vector-treated (PBP8 wt-CsgA DSS$^+$), and PBP8 with CsgA-TFF3 vector-treated (PBP8 CsgA-TFF3 DSS$^+$) with $n = 4$–5 in each group for each set of experiments. During the course of one experiment (14 days), mice were fed with normal mouse chow ad libitum while animal body weight and water intake were evaluated daily. At the beginning of the experiment (day-3), mice in all groups started receiving 10 g L$^{-1}$ L-(+)-arabinose and 1 g L$^{-1}$ kanamycin in drinking water. Meanwhile, the PBP8 bacteria cultures were grown, induced overnight, centrifuged, and resuspended to an OD$_{600}$ of 10 in 20% sucrose in PBS. Then, mice received 100 μL PBS (in PBS DSS$^-$ and PBS DSS$^+$ groups), PBP8, PBP8 wt-CsgA, and PBP8 CsgA-TFF3 by rectal administration once daily throughout the experimental period. Three days after the start of bacterial administration (day 0), colitis was induced by the addition of DSS (MW 40,000, Alfa Aesar) to a final concentration of 3% in the drinking water that also contained L-(+)-arabinose and kanamycin. Mice in all groups except the non-colitic (PBS DSS$^-$) group received DSS treatment for 5 days. After DSS removal (day 6), all mice were given the L-(+)-arabinose and kanamycin water until day 10 when they were sacrificed.

In an additional experiment to assess the PBP8 N22-TFF3 control, mice were randomly assigned to three groups: colitic (PBS DSS$^+$), PBP8 with CsgA-TFF3 vector-treated (PBP8 CsgA-TFF3 DSS$^+$), and PBP8 with N22-TFF3 vector-treated (PBP8 N22-TFF3 DSS$^+$) with $n = 6$–8 in each group. The DSS-induced colitis experiment was performed in a similar manner to the experiment above.

The colon of each mouse was removed and its length were measured. The feces were gently scraped off. The distal colon of each mouse was divided into three sections (about 1 cm each). The most proximal section was placed in RNAlater solution, frozen with liquid nitrogen, and stored at −80 °C until RNA extraction. The middle section was weighed, frozen with liquid nitrogen and stored at −80 °C for protein quantification. The most distal section was fixed in 4% paraformaldehyde in PBS buffer overnight at 4 °C for histological analysis. Detailed protocols for the determination of disease activity and histological studies can be found in additional materials and methods.

**Determination of disease activity**. In addition to the body weight, each animal was monitored daily for the presence of blood in the fecal samples and the stool consistency. The parameter and its corresponding scoring that constituted the disease activity index (DAI), shown in Table 1, was adapted from a published protocol[50,67]. The DAI was calculated by addition of scores from all three parameters.

**Histological studies**. The fixed colonic samples were embedded in paraffin, sectioned (5 μm) and stained with hematoxylin and eosin. The sections were scored blindly by a pathologist for histological evidence of intestinal damage by DSS with a scoring system described previously[45–47]. In brief, the sections were evaluated for the amount and depth of inflammation using a 0 to 3 score range and the extent of crypt damage using a 0 to 4 score range. Each feature got multiplied by the percentage of the area involved as indicated in Table 2 and added to the total summation. The maximum possible score is 40.

**Luminex multiplex immunoassay**. Five-hundred microliters of 1x mammalian cell lysis buffer and 5 mm-stainless steel beads were added to frozen tissue samples in 2 mL microcentrifuge tubes. The samples were then homogenized using Tissue-Lyser LT at 50 Hz for 10 min at 4 °C. Following homogenization, the samples were centrifuged at $10,000 \times g$ for 10 min at 4 °C and the supernatants were transferred to new sample tubes. The samples were tested for six cytokines: IFN-γ, IL-1β, IL-6, IL-10, IL-17A, and TNF-α, using a Bio-Plex Pro Mouse Cytokine Th17 Panel A 6-Plex kit in accordance with the manufacturer's protocol. The final 96-well plate was processed using the Bio-Plex 3D system and the concentration of each cytokine was determined using Bio-Plex Manager software.

**Gene expression analysis by qRT-PCR**. RNAlater-stabilized tissues were subjected to total RNA extraction using the TissueLyser LT and RNeasy plus mini kit in accordance with the manufacturer's protocol. RNA samples were eluted with 100 μL RNase free water provided from the kit. Following the elution, the concentration of RNA was determined by spectroscopy using the Nanodrop 2000c. Ten nanograms of RNA was analyzed using specific primers for each gene of interest (Supplementary Table 5) and a KAPA SYBR FAST One-Step qRT-PCR kit with CFX96 real-time PCR detection system in accordance with the manufacturer's protocol. The Pfaffl method was used to normalize the expression result[68]. In brief, the $E^{-\Delta\Delta Ct}$ value, where $E$ was the primer efficiency of each primer pair and $-\Delta\Delta Ct$ was difference in the average Ct value of the control groups and the sample, was used to transform the Ct values of each sample into the expression values. Then, the expression values of the housekeeping gene, glyceraldehyde-3-phosphate dehydrogenase (GAPDH), were used to normalize the expression values of each gene in this study.

**Statistics**. The investigators were not blinded to the experimental conditions during experiments and outcome assessment, except for the pathohistology analysis. Data are presented as the arithmetic means plus or minus SEM. The data were analyzed using GraphPad Prism 7. The statistical significance of the bar and scatter bar plots were determined using one-way ANOVA followed by Dunnett's (Fig. 2 and Supplementary Fig. 1), Tukey's (Fig. 3 and Supplementary Fig. 2) or Fischer's LSD multiple comparison (Figs. 4 and 6d, e, k, l and Supplementary Figs. 7–9). The time course experiments, such as percentage weight change and DAI, were analyzed using two-way ANOVA following by Dunnett's multiple comparison. An associated probability ($p$-value) of <0.05 was considered significant and given one star, whereas the $p < 0.01$ was given two stars, $p < 0.001$ was given three stars, $p < 0.0001$ was given four stars, and $p > 0.05$ was given "ns" accordingly.

**Reporting summary**. Further information on research design is available in the Nature Research Reporting Summary linked to this article.

## Data availability
The authors declare that all relevant data supporting the findings of this study and the plasmids and strains used are available within the article and its supplementary files or from the corresponding authors on request. Source data are provided in the Source Data file.

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

## Acknowledgements

This work made use of the Harvard Digestive Diseases Center (HDDC), the Harvard Center for Nanoscale Systems (CNS) and Harvard Medical School ICCB-Longwood Screening Facility, and the Wyss Institute for Biologically Inspired Engineering. We would like to thank Trevor R. Nash, Frederick R. Ward, Amanda Graveline, Andyna Vernet, Frank Urena, Jessica J. Kim, Daniel Um, Mofeyifoluwa Edun, Frederic Vigneault, Elaine Lim, Thomas Ferrante, Garry Cuneo, Magdalena Kasendra, and Rachelle Prantil-Baun for their help. P.P. thankfully acknowledges the royal Thai government scholarship. D.B.C. gratefully acknowledges the National Institutes of Health grant (2T32CA009216-36). This work was supported by National Institutes of Health (1R01DK110770-01A1), the Blavatnik Biomedical Accelerator fund, and the Wyss Institute for Biologically Inspired Engineering.

## Author contributions

P.P., A.D.-T., N.S.J. conceived the idea and designed the experiments. P.P. cloned CsgA variant plasmids. P.P. performed the curli characterization experiments and data analysis. P.P. and F.B. performed the in vitro tissue culture pathogenicity experiments and data analysis. P.P., A.D.-T., I.G. performed the DSS colitis mouse experiments and sample collections, and data analysis. P.P. performed the qRT-PCR and Luminex, and data analysis. D.B.C. performed the blinded histopathology scoring and P.P. performed the data analysis. P.P., A.D.-T., and I.G. performed the residence time experiment. P.P. and A.D.-T. performed the IVIS experiment. P.P. performed the fecal filtration ELISA and curli immunohistochemical experiment. The manuscript was written through contributions of all authors. The final version of the manuscript has been approved by all the authors.

## Competing interests

The authors declare no competing interests.
