## [Peer Review File · Nature Communications]

Reviewers' Comments:

Reviewer #1:

Remarks to the Author:

Manuscript by Praveschotinunt et al investigates the role of a genetically engineered *E. coli* strain that produces trefoil factors on curli fibers in the intestinal tract using a mouse model for IBD. Although the idea is nice and novel there are few major issues with the manuscript. Curli fibers were shown previously to interact with epithelial cells and dampen inflammation and improve barrier function in a mouse model of IBD (Oppong et al 2013 *Infection and Immunity*, Oppong et al 2015 *npj Biofilms and Microbiomes*). The manuscript in its current form is missing many controls to confirm that the effects seen by the engineered strains that express CsgA-TFF constructs are not due to CsgA or curli by itself. Additional experiments are also recommended to establish the superior activity of CsgA-TFF constructs compared to CsgA or TFF alone. In its current form, the manuscript does not convince this reviewer on the effect of the engineered strains.

Figure 2A. It is not clear from the figure or the text whether the EcN produces curli fibers in this figure. If this is a EcN delta csg mutant that expresses the constructs, please label the figures correctly. Otherwise, the EcN normally produces curli and binds to CR. Explain why there is not CR binding in this figure.

Figure 2. This comment goes to the whole figure. There is not enough information regarding the experimental conditions for this figure. What are the conditions that the bacteria were grown in? Are the TFF1-2-3 are stimulated by arabinose? What are the background levels for the vector only controls for Figure 2A and 2B

Figure 3. *S. Typhimurium* and *S. Typhi* are different serotypes. If you used *S. typhimurium* SL1344, the label on the graphs should read *S. typhimurium*.

Figure 4C. the ELISA should be expressed in pg/ml not ODs. There are strains missing from the ELISA. All the samples should be included. A positive control should be used to determine the amount of curli in samples.

Figure 4. This is in vivo expression not in situ.

Figure 5. It is hard to tell the lumen and the tissue from these samples. There is no evidence that the 6X His staining is specific to bacteria. This could be an artifact. Appropriate positive and negative controls should be included. For instance staining with the EcN delta csg strain to show that there is no unspecific staining. Also EcN could be stained with LPS antibodies to show the localization of the bacteria.

Figure 6. Why did the authors choose to administer bacteria rectally. They used oral colonization in Figure 4. However they change their administration protocol and use rectal administration. DSS-colitis model affects the whole intestinal tract. Therefore it is usually acceptable that the administration of the microbes or drugs are done orally. Do they expect the oral administration will give the similar result? If so, why wasn't this route chosen?

Figure 6D-E. Curli by itself was shown to ameliorate colitis. There is not difference in the colon length and the histology scores between the groups that receive PBP8 CsgA and PBP8 CsgA-TFF3. How does the authors conclude an effect of TFF8 while this effect could be just CsgA by itself.

Fig 7A. The *tjp1* expression is referred to be high in the results while the difference is 2-fold expression difference. These are not high levels but for tight junction proteins it is usually hard to dissect protein levels using qPCR. If authors want to make claims on barrier integrity and TJ proteins, they should use WB or immunostaining to strengthen their results. Again there is not significant difference between PBP8 CsgA and PBP8 CsgA-TFF3 groups.

Figure 7. Overall the statistics for this figure should be acknowledged. It is not clear what kind of analysis was done between groups

These complicated assays can be done using Caco-2 cells and see if the bacterial constructs improve barrier function. The tight junction proteins could also be examined in this clean setting.

I recommend to add a figure to show that TFFs augment barrier function or anti-inflammatory effect. They could be compared to curli to establish the impact of the engineered strains.

Reviewer #2:

Remarks to the Author:

Herein, Praveschotinunt and colleagues report the generation of a genetically-engineered *Escherichia coli* Nissle 1917 that creates a fibrous matrix displaying trefoil factors in vitro and in vivo. The rectal (but not oral) route of administration of such novel strain of probiotic efficiently improved disease severity in a preclinical model of colitis that is induced by dextran sodium sulfate. The authors correctly cited works that already demonstrated impacts of *Escherichia coli* Nissle 1917 in some IBD patients. In general, this is an interesting study, but lacks mechanistic depth. For example, the authors state that engineered curli fibers do not confer pathogenicity to EcN in vitro, but it remains elusive whether this will still be valid in the context of dysbiosis and particularly in the context of a lowered abundance of mucus-degrading bacteria. Additional preclinical studies including hypothesis driven experimentation in mice that are spontaneously prone to develop colitis should be employed to rigorously test this hypothesis. Foremost, it probably is not reasonable to make statements on IBD using a single mouse model of colitis and at only one endpoint. Additional models should be employed such as oxalozone or TNBS induced injury to assess the contribution of T cells. In addition, it remains elusive whether the beneficial effect of this genetically-engineered *Escherichia coli* Nissle 1917 may persist in the context of dysbiosis and over time in a chronic model of colitis. Indeed, no attempt was made to test whether neutralizing antibodies may be produced in response to the trefoil factors that are produced by this probiotic. As minor comments, I would recommend thorough proofing for grammatical and typographical errors.

Reviewer #3:

Remarks to the Author:

In this work, the authors demonstrated an engineered probiotics Nissle 1917 (EcN) to alleviate symptoms in a dextran sodium sulfate-induced colitis mouse model. Specifically, EcN was engineered to produce a fusion protein, synthetic curli-trefoil factor 3 (CsgA-TFF3), which was then assembled with other EcN-derived curli fibers to form an anti-inflammatory fibrous matrix. Then they conducted in vitro and in vivo experiments to examine the efficacy of the engineered EcN. They showed that the deletion of curli operon from EcN genome and the subsequent overexpression of the curli fibers did not change the pathogenicity of EcN. The authors also attempted to understand the mechanism behind the anti-inflammatory effect of the synthetic fibrous matrix and confirmed a reduction in Th17 responses upon treatment with the engineered EcN. Overall, the manuscript is well written, and the data provided are sufficient to support the conclusions made. The findings would provide valuable insights into the development of therapeutics on IBD using the engineered EcN strains. The following comments should be addressed during revision.

Major comments:

1. Figures 4 and 5: It is unclear whether PBP8 strain or EcN strain was used. Please clarify whether the current conclusion made according to Figures 4 and 5 is consistent with that when the native

curli operon is present in EcN strain.

2. Figure 5: For clarity and accuracy, this figure needs improvement. Specifically, a bright field image for each sample should be provided, which is necessary to identify the artefacts in the images due to luminal contents. Furthermore, the position of the epithelium/mucus/lumen layers should be indicated or specified in the images. Quantitative data such as fluorescence intensity should be included to further support the conclusion made.
3. Figure 6C and line 280: A disease activity index (DAI) was used to monitor the prognosis in mice. I recommend providing the citation for this method. Otherwise, the authors should clarify how DAI can be used to evaluate the prognosis.
4. Figures 5, 6 and 7: The three figures were used to compare the resistance conferred by probiotic strains expressing CsgA-TFF3. The strain expressing CsgA was used as a control. Another control group of the EcN strain expressing TFF3 alone should be also included. The subsequent results obtained should be discussed accordingly in the text.
5. Fig 7F: For readers to better understand the results, I recommend discussing the reason why the level of TNF-alpha expression in the PBS DSS- group is the highest.
6. To justify the significance of this work, I recommend comparing and discussing the improvement observed for IBD treatment by PATCH over other methods, e.g. 5-ASA. Also, would the multi-pronged treatment described in the introduction work better than PATCH?
7. Figures 4A and 6B: Different strains were compared. For clarity, I recommend explaining the difference observed in the strains compared.
8. In this study, changes in the expression of IL-6, IL-17A and TNF-alpha were studied. I recommend studying the changes in expression of other cytokines, such as IL-4, IL-9 and IL-13, which are possibly regulated by the pathogenesis of IBD- or DSS-induced colitis.
9. A recent study (<https://www.ncbi.nlm.nih.gov/pubmed/28627600>), microRNA-7-5p (miR-7-5p) was identified as a negative regulator of TFF3 and found to be upregulated in lesion tissues. Is a similar regulator involved in the regulation of TFF3 upon PATCH treatment?
10. In this study, the plasmid with antibiotic resistance was used to express CsgA-TFF3. As mentioned in "section" discussion, a second generation PATCH system will be developed by using an antibiotic-free plasmid system. I recommend discussing the strategy of (1) expressing csgA-TFF3 upon induction by inducers which are more relevant to the microenvironment of IBD rather than arabinose, and (2) integrating the csgA-TFF3 cassette into the genome, which may overcome the potential issue on instability of the plasmid and CsgA-TFF3 expression.

Minor comments:

1. The title of the manuscript is not informative enough. I recommend providing a more descriptive title.
2. Figures 2C-G and 6F-J: For clarity, please provide labels or names e.g. GFP, wt-CsgA in each image instead of using characters, e.g. C, D, etc.
3. Figures 3: Please change "S. Typhi" in X-axis to italic "S. typhi", and adjust the colour scheme of the bars in Figure 3D to be consistent with those in Figure 3A-C. Furthermore, please make the font of "S. typhi" in X-axis consistent.
4. Figure 6D: The exact p-values are recommended for the ns dataset.
5. Line 205: CFU should be presented using CFU/g faecal sample.
6. Line 267: Please rephrase "over several days" to be more specific, e.g. "over five days".
7. In section Materials and Methods: Please change "Arabinose" to "arabinose".
8. Line 335: Please change "This could the explained..." to "This should be explained".
9. Line 338: Please change "form" to "from".
10. Line 541: The additional space should be removed in "3% ".

We thank all the reviewers for their constructive feedback, which we have used to improve the quality of our manuscript. Below are point-by-point responses to the Reviewer critiques.

Original Reviewer critiques appear in black

Our responses to the critiques appear in blue

Quoted edits to the main text appear in blue italics. Line numbers refer to the revised manuscript.

Reviewers' comments:

Reviewer #1 (Remarks to the Author):

Manuscript by Praveschotinunt et al investigates the role of a genetically engineered *E. coli* strain that produces trefoil factors on curli fibers in the intestinal tract using a mouse model for IBD. Although the idea is nice and novel there are few major issues with the manuscript. Curli fibers were shown previously to interact with epithelial cells and dampen inflammation and improve barrier function in a mouse model of IBD (Oppong et al 2013 *Infection and Immunity*, Oppong et al 2015 *npj Biofilms and Microbiomes*). The manuscript in its current form is missing many controls to confirm that the effects seen by the engineered strains that express CsgA-TFF constructs are not due to CsgA or curli by itself. Additional experiments are also recommended to establish the superior activity of CsgA-TFF constructs compared to CsgA or TFF alone. In its current form, the manuscript does not convince this reviewer on the effect of the engineered strains.

We thank Reviewer 1 for providing constructive feedback that could improve the quality of our manuscript. We agree that the effects of the curli fiber backbone, composed of CsgA units, could contribute to the anti-inflammatory effects of the treatment. We acknowledge this in the manuscript (line 451-3), but we have added another sentence with references to the Discussion section in order to clarify further:

"In fact, wild-type curli fibers were shown to have some anti-inflammatory and barrier protective properties, which could confound our ability to observe the effect of the appended trefoil factors^{26,46,50,56}(in text)"

Point-by-point responses to the other critiques from Reviewer 1 are below.

1. Figure 2A. It is not clear from the figure or the text whether the EcN produces curli fibers in this figure. If this is a EcN delta csg mutant that expresses the constructs, please label the figures correctly. Otherwise, the EcN normally produces curli and binds to CR. Explain why there is not CR binding in this figure.

The data labeled "EcN" in this figure comes from a strain of *E. coli* Nissle 1917 called "Prop-Luc" that contains the native, chromosomal copies of the curli genes, and also a chromosomal insertion of the lux operon for tracking purposes. The strain is also carrying a plasmid encoding GFP as an alternative to a "vector only" control. Because this strain harbors chromosomal curli genes, and for the sake of simplicity, we refer to this strain as EcN. Throughout the manuscript, we label EcN delta csg as the "PBP8" strain. We are aware that many people have reported that EcN produces curli fibers prolifically¹⁻³, but such data are obtained using expression at room temperature, and sometimes in minimal media – conditions that promote curli production. In this experiment, we grew and induced the expression of the synthetic curli operon in high osmolarity media (LB Broth) and at 37°C. Under these conditions, the chromosomally-encoded curli operon in EcN should be repressed through the OmpR-mediated down-regulation of *csgD*¹. Therefore, the fact that we do not observe curli production from EcN under these conditions is expected. This aspect of curli gene regulation was discussed in the original manuscript (Lines 155-7 in the revision):

"The inclusion of the other genes of the curli operon was necessary to increase secretion efficiency, because the curli genes in the EcN chromosome are downregulated at physiological temperature and osmolarity."

We also have unpublished data showing that EcN grown in YESCA media (low salt) and 28°C do produce curli fibers, as measured by CR binding. We thought it would be redundant to include this data, given the published accounts, but we can include it if necessary.

2. Figure 2. This comment goes to the whole figure. There is not enough information regarding the experimental conditions for this figure. What are the conditions that the bacteria were grown in? Are the TFF1-2-3 are stimulated by arabinose? What are the background levels for the vector only controls for Figure 2A and 2B

All the bacteria in Figure 2 were grown in LB media and expression was induced by the addition of arabinose during logarithmic phase. Although the expression conditions are described fully in the Methods section, we have included them in the revised figure caption as well to address the reviewer's concerns. The background levels of curli production observed in "vector only" controls are already represented in the figure, labeled as "EcN GFP". As described above, this is actually the "Prop-Luc" strain of EcN harboring vector that encodes for GFP in place of the curli genes. We have modified the main text as following to further emphasize the fact.

"In order to confirm that curli fibers decorated with TFFs could be produced by EcN, as they can in laboratory strains of E. coli^{22(in text)}, we transformed EcN with the panel of synthetic curli plasmid constructs, in addition to a vector encoding GFP in place of the curli genes as a negative control. The transformed cells were cultured at 37°C in high osmolarity media to mimic physiological conditions and induced with L-(+)-arabinose. A quantitative Congo Red binding (CR) assay, normally used for curli fiber detection^{22,25(in text)}, indicated that wild-type CsgA and all three CsgA-TFF fusions could be expressed and assembled into curli fibers under physiological conditions, while EcN with the GFP-expressing control vector showed no CR binding (Figure 2A)."

We have also moved the methods section on "In vitro expression of engineered curli fiber" from the supplementary materials and methods to the main text's materials and methods to better emphasize the expression scheme.

3. Figure 3. S. Typhimurium and S. Typhi are different serotypes. If you used S. typhimurium SL1344, the label on the graphs should read S. typhimurium.

We thank the reviewer for catching this typo. We have changed the label on figure 3 and S2 from S. typhi to S. typhimurium.

4. Figure 4C. the ELISA should be expressed in pg/ml not ODs. There are strains missing from the ELISA. All the samples should be included. A positive control should be used to determine the amount of curli in samples.

We have updated the figure to show all the samples. In terms of the y-axis units, we agree with the Reviewer's suggestion that absolute units are, in general, a better way to report ELISA data. However, the inclusion of a "positive control", as suggested by the Reviewer, is incompatible with this assay, which uses a 0.2 µm filter to separate particulate-bound, insoluble curli fibers from soluble components of the homogenized fecal samples. In our experience, purified and reassembled CsgA is difficult to dilute reliably due its insolubility and the in vitro reassembled curli fibers are not consistently retained by the filter in such an assay, making the conversion to absolute units challenging, if not impossible. We feel that the relative units for this ELISA adequately support the main point of the figure, which is that engineered EcN strains that are supposed to produce curli fibers lead to higher signals in the ELISA per unit mass of homogenized fecal samples.

5. Figure 4. This is in vivo expression not in situ.

We changed the text from *in situ* to *in vivo*

6. Figure 5. It is hard to tell the lumen and the tissue from these samples. There is no evidence that the 6X His staining is specific to bacteria. This could be an artifact. Appropriate positive and negative controls should be included. For instance staining with the EcN delta csg strain to show that there is no unspecific staining. Also EcN could be stained with LPS antibodies to show the localization of the bacteria.

We have performed the LPS staining using the anti-E. coli LPS antibody and replaced figure 5 with the fluorescence images from the new immunostaining. We also have added the requested control, which is the EcN with no CsgA plasmid. We did not use the PBP8 strain in this study but EcN control should work just as well because EcN does not express native curli under physiological conditions. In this experiment, the anti-LPS staining revealed that EcN, EcN wt-CsgA and EcN CsgA-TFFs could be observed throughout the gut lumen and near the most superficial layers of mucus, while the PBS control showed minimal background staining.

7. Figure 6. Why did the authors choose to administer bacteria rectally. They used oral colonization in Figure 4. However they change their administration protocol and use rectal administration. DSS-colitis model effects the whole intestinal tract. Therefore it is usually acceptable that the administration of the microbes or drugs are done orally. Do they expect the oral administration will give the similar result? Is so, why wasn't this route chosen?

We addressed the rationale for the rectal administration route in the original text (Line 268 in revised manuscript):

"Pilot experiments with oral administration of PBP8 strains were not very effective in decreasing disease symptoms. However, histological analysis and further literature consultation revealed that DSS induced colitis was most severe in the distal colon, whereas the engineered bacteria resided mostly in the cecum and proximal colon (Figure S5)^{24,37,38}(in text). In order to circumvent this peculiarity of the murine DSS model and investigate the efficacy of our approach, we pivoted to rectal administration of the bacteria so that they could easily co-localize with the affected tissues. Notably, we do not envision that this issue would affect the efficacy of engineered bacteria in other models or in humans, as both oral and rectal deliveries are viable routes of drug administration depending on the patient's disease localization."

We respectfully disagree with the reviewer's statement that the DSS model affects the whole GI tract. Although the manner in which the DSS-induced damage presents may vary based on the details of the DSS protocol, in our experience, the damage was highly localized to the distal colon. The references we cite also support the distal localization of the DSS-induced damage. We think that this is a peculiarity of the particular model we are using. In humans and other models, we anticipate that the inflammation and bacteria would co-localize after oral administration, although this likely also varies on a patient-by-patient basis. In either case, we consider both oral and rectal administration routes clinically relevant delivery methods.

8. Figure 6D-E. Curli by itself was shown to ameliorate colitis. There is not difference in the colon length and the histology scores between the groups that receive PBP8 CsgA and PBP8 CsgA-TFF3. How does the authors conclude an effect of TFF8 while this effect could be just CsgA by itself.

We recognize that the curli fiber backbone has its own documented interactions with the gut epithelium, some of which can enhance barrier function. We acknowledged this in the original manuscript (Line 395 in the revised manuscript):

"We also recognize that the curli fibers themselves are not a "blank slate" material in that CsgA already has numerous known interactions with host cells and tissues that could confound the effects of the displayed domains"

Overall, we see this as a positive aspect of our approach, since the action of the curli fibers themselves and the displayed domains could act synergistically to reinforce the epithelial barrier. As for the difference between wt-CsgA fibers and CsgA-TFF3 fibers, we acknowledge that the cytokine analysis from tissue samples and the histology scoring did not show a statistically significant difference between these two conditions. However, the weight loss curves and disease activity index (DAI) in Figure 6B,C did show clear differences between these conditions. We think that the significance of the histology scoring may have been lessened by two factors. First, at the experimental endpoint the animals from all conditions had been allowed a 5-day recovery period after the cessation of DSS treatment, which could have lessened any observable differences between the samples across conditions. Second, the histology scoring we reported was taken from colonic cross sections, which may not have accurately represented the colon as a whole.

We have also amended the discussion of the paper to address this issue, as following:

"While this may or may not impede further development of PATCH with curli fibers as a scaffold, we know that the biosynthetic machinery dedicated to curli secretion can tolerate a wide range of heterologous proteins^{57(in text)}. We are therefore in the process of exploring other combinations of scaffolding proteins and bioactive domains that can be secreted through the curli (a.k.a. "Type VIII") pathway to circumvent these confounding effects and probe different therapeutic modalities."

9. Fig 7A. The tjp1 expression is referred to be high in the results while the difference is 2-fold expression difference. These are not high levels but for tight junction proteins it is usually hard to dissect protein levels using qPCR. If authors want to make claims on barrier integrity and TJ proteins, they should use WB or immunostaining to strengthen their results. Again there is not significant difference between PBP8 CsgA and PBP8 CsgA-TFF3 groups.

Although there are some studies that use qPCR to monitor tjp-1 expression in colonic tissues⁴, brain endothelial cells⁵, and human breast cancer cells⁶, we agree that there might be better methods. In terms of the difference between the wt-CsgA and CsgA-TFF3 group, we acknowledged the fact that CsgA has some overlapping anti-inflammatory property and we have mentioned that in the text as specified in the response above. In light of these factors, we decided to move the part of the figure involving tjp-1 to the supplementary document.

10. Figure 7. Overall the statistics for this figure should be acknowledged. It is not clear what kind of analysis was done between groups

In an effort to conserve space in the figure captions, we summarized the statistical analysis for the data in every figure in the *Statistics* sub-section within materials and methods section.

11. These complicated assays can be done using Caco-2 cells and see if the bacterial constructs improve barrier function. The tight junction proteins could also be examined in this clean setting.

We thank the reviewer for the suggestion. We performed additional experiments examining the tight junction response in polarized Caco-2 monolayers using TEER measurements. We treated the cells with IFN- γ and TNF- α while incubating with purified wild-type curli or curli TFF3 as well as soluble TFF3. We then compared the TEER values at 24 hours post-inoculation with the TEER values before treatment in order to observe the prophylactic effects of wild-type curli versus curli-TFF3 (Figure S8). The results showed that wt-CsgA, soluble TFF3, and CsgA-TFF3 all decreased the drop in TEER values to roughly the same extent. Experiments using fluorescently labelled

dextrans showed similar results and immunostaining for proteins associated with tight junctions was inconclusive as a qualitative measure of barrier integrity. We have included the TEER data as a supplementary figure in the revised manuscript and edited the main text accordingly. Given the inherent difficulty in disentangling the effects of wild-type curli fibers and TFF3, we feel that rigorous experiments to probe the mechanism of the effects we observe are outside the scope of this paper. Future experiments could use TLR2 or TLR2/1 knockout models in both *in vitro* and *in vivo*. Nevertheless, the other characterization techniques we employ (weight change, DAI, colon length, IL-17A) clearly show a significant difference between the effects of wt-CsgA and CsgA-TFF3.

12. I recommend to add a figure to show that TFFs augment barrier function or anti-inflammatory effect. They could be compared to curli to establish the impact of the engineered strains.

We feel that the weight loss and DAI data show conclusively that the wt-CsgA fibers and the CsgA-TFF fibers lead to different outcomes *in vivo*. We agree that the cytokine analysis and histology do not reveal the origin of these differences quite as well. However, we believe that adding the study section just for the soluble TFFs alone would be outside the scope of this study because there are already quite a few studies focusing on the effects of TFFs alone in multiple models showing their barrier protective effects⁷⁻⁹.

Reviewer #2 (Remarks to the Author):

Herein, Praveschotinunt and colleagues report the generation of a genetically-engineered *Escherichia coli* Nissle 1917 that creates a fibrous matrix displaying trefoil factors *in vitro* and *in vivo*. The rectal (but not oral) route of administration of such novel strain of probiotic efficiently improved disease severity in a preclinical model of colitis that is induced by dextran sodium sulfate. The authors correctly cited works that already demonstrated impacts of *Escherichia coli* Nissle 1917 in some IBD patients. In general, this is an interesting study, but lacks mechanistic depth.

We thank Reviewer 2 for their constructive comments. We have addressed them below in a point-by-point response and we have also revised the manuscript accordingly.

1. For example, the authors state that engineered curli fibers do not confer pathogenicity to EcN *in vitro*, but it remains elusive whether this will still be valid in the context of dysbiosis and particularly in the context of a lowered abundance of mucus-degrading bacteria.

It is somewhat unclear what the reviewer means by dysbiosis in this context. We have run extensive experiments in which we administer the engineered bacteria to otherwise healthy mice that are receiving antibiotics (kanamycin). The antibiotic is part of the experiment in order to ensure that the EcN maintains the plasmid, but it also leads to dysbiosis, in that it kills a range of other enteric bacteria. Some of these experiments are shown in Figure 4A, and others are pilot studies that were not included in the manuscript. In none of these experiments did we observe any morbidity (e.g. weight loss, sluggishness) that would be associated with pathogenicity from any of the administered EcN strains, even over the course of 25-30 days with repeated administrations. The DSS model represents another scenario in which dysbiosis occurs. In our experiments (Figure 6B,C), we did not observe any worsening of weight loss or other physiological indications of bacterial pathogenicity when comparing mice that received DSS with no bacteria and mice that received both DSS and bacteria. In fact, the CsgA-TFF3 expressing strain ameliorated weight loss and DAI. We feel that these data are sufficient evidence that the engineered strains are not pathogenic, for the purposes of our manuscript.

2. Additional preclinical studies including hypothesis driven experimentation in mice that are spontaneously prone to develop colitis should be employed to rigorously test this hypothesis.

Foremost, it probably is not reasonable to make statements on IBD using a single mouse model of colitis and at only one endpoint. Additional models should be employed such as oxalozone or TNBS induced injury to assess the contribution of T cells.

Given the limitations of any of the existing preclinical models for IBD, we recognize the need for demonstrating efficacy in multiple models before reaching a conclusion that our therapeutic strategy may be effective against the disease. After some effort to perform supplementary experiments with the TNBS colitis model, we found that the model itself required more optimization in our hands before we could use it meaningfully to corroborate our results with the DSS model, and that the optimization would take longer than we have for the resubmission of this manuscript. Nevertheless, we feel the existing data still strongly supports the claims that 1) the PATCH system is a new way to think about delivering therapeutic proteins to the mammalian gut with a self-assembling matrix material, rather than soluble drugs, and 2) the CsgA-TFF3 fusion ameliorates the severity of the DSS treatment. We have revised the main text of the manuscript to remove any suggestion that the CsgA-TFF3 system would be effective in treating IBD, and have refocused on its potential role in promoting mucosal wound healing and reinforcing barrier function.

3. In addition, it remains elusive whether the beneficial effect of this genetically-engineered *Escherichia coli* Nissle 1917 may persist in the context of dysbiosis and over time in a chronic model of colitis.

We agree that additional preclinical models of IBD would be needed in order to de-risk clinical translation, although none of them accurately represent human disease, which has multifactorial causes and is affected greatly by environmental factors. We feel that such models should be used strategically to probe specific mechanisms of action. In our study, the primary hypothesis about how the TFFs exert a biological effect is by stimulating mucosal healing and reinforcing barrier function. We think the DSS model, which causes the acute injury to the colon epithelium is an appropriate model for evaluating our system. Other models tend to involve inflammation caused by immune cell activation which the TFFs might not have strong effect against. We feel that additional disease models would be beyond the scope of this manuscript.

4. Indeed, no attempt was made to test whether neutralizing antibodies may be produced in response to the trefoil factors that are produced by this probiotic.

We acknowledge that neutralizing antibodies could be a problem for any protein-based therapeutic. However, we feel that experiments to identify their presence would be beyond the scope of this manuscript. We present several other papers, published in journals of stature comparable to *Nature Communications*, as examples of anti-inflammatory therapies targeted to the gut that chose to leave the investigation of neutralizing antibodies to future studies⁹⁻¹¹.

5. As minor comments, I would recommend thorough proofing for grammatical and typographical errors.

We have thoroughly proof-read the revised manuscript and addressed any typographical errors that we found.

Reviewer #3 (Remarks to the Author):

In this work, the authors demonstrated an engineered probiotics Nissle 1917 (EcN) to alleviate symptoms in a dextran sodium sulfate-induced colitis mouse model. Specifically, EcN was engineered to produce a fusion protein, synthetic curli-trefoil factor 3 (CsgA-TFF3), which was then assembled with other EcN-derived curli fibers to form an anti-inflammatory fibrous matrix. Then they conducted in vitro and in vivo experiments to examine the efficacy of the engineered EcN. They showed that the deletion of curli operon from EcN genome and the subsequent overexpression of the curli fibers did

not change the pathogenicity of EcN. The authors also attempted to understand the mechanism behind the anti-inflammatory effect of the synthetic fibrous matrix and confirmed a reduction in Th17 responses upon treatment with the engineered EcN. Overall, the manuscript is well written, and the data provided are sufficient to support the conclusions made. The findings would provide valuable insights into the development of therapeutics on IBD using the engineered EcN strains. The following comments should be addressed during revision.

We thank Reviewer 3 for their constructive comments. The critiques are addressed in the revised manuscript and the point-by-point response below.

Major comments:

1. Figures 4 and 5: It is unclear whether PBP8 strain or EcN strain was used. Please clarify whether the current conclusion made according to Figures 4 and 5 is consistent with that when the native curli operon is present in EcN strain.

The data presented in Figures 4 and 5 make use of the Prop-Luc strain of *E. coli* Nissle. This strain contains a chromosomal insertion of a lux operon to aid with bacterial tracking. It also contains the native chromosomal copies of the curli genes. Since the native curli genes are not expressed under physiological conditions¹, as we addressed in our response to Reviewer 1 above, and confirmed with our experiments shown in Figures 2A-C and 4C, they should not interfere with the effects of EcN *in vivo*. We have updated the labeling of data in the figures and captions, and amended the main text to clarify these points.

2. Figure 5: For clarity and accuracy, this figure needs improvement. Specifically, a bright field image for each sample should be provided, which is necessary to identify the artefacts in the images due to luminal contents. Furthermore, the position of the epithelium/mucus/lumen layers should be indicated or specified in the images. Quantitative data such as fluorescence intensity should be included to further support the conclusion made.

We have included an updated figure with better labeling of the epithelial tissue and gut lumen, in addition to staining with other antibodies (e.g. anti-LPS) that shows less background staining compared to the original anti-His6 staining. We have also included brightfield images to depict the gut tissues versus the gut luminal contents.

3. Figure 6C and line 280: A disease activity index (DAI) was used to monitor the prognosis in mice. I recommend providing the citation for this method. Otherwise, the authors should clarify how DAI can be used to evaluate the prognosis.

We have moved the details and citation of the DAI protocol from the supplementary materials and methods to the main text to make the protocol more accessible to the reader.

4. Figures 5, 6 and 7: The three figures were used to compare the resistance conferred by probiotic strains expressing CsgA-TFF3. The strain expressing CsgA was used as a control. Another control group of the EcN strain expressing TFF3 alone should be also included. The subsequent results obtained should be discussed accordingly in the text.

We agree with the reviewer that the secretion of soluble TFF3 from the engineered EcN strains would be an interesting comparison. However, this requires more time to develop than we have for the resubmission of the manuscript. Our preliminary experiments show that the soluble TFF3 can be secreted from the engineered EcN, but performing a head-to-head comparison complicated by difficulties in comparing the concentration of soluble TFF3 to curli-bound TFF3. Therefore, we feel it is outside the scope of this revision.

5. Fig 7F: For readers to better understand the results, I recommend discussing the reason why the level of TNF-alpha expression in the PBS DSS- group is the highest.

This result was also surprising to us. Given the high variability across samples, the assay may have been near its detection limit for TNF-alpha. In the revised manuscript, we have removed this data so as not to cause confusion.

6. To justify the significance of this work, I recommend comparing and discussing the improvement observed for IBD treatment by PATCH over other methods, e.g. 5-ASA. Also, would the multi-pronged treatment described in the introduction work better than PATCH?

We agree that the multi-pronged treatment has the potential to be even better than PATCH treatment alone. However, as this is the first demonstration of PATCH as a therapeutic approach, we feel that combining it with another therapy could confound the results. As such, we felt it would be more appropriate for a follow-up study.

Nevertheless, we reassessed the literature on the use of 5-ASA in the DSS colitis murine model^{12,13}. Based on these works, we did not find 5-ASA to be effective in terms of preventing weight loss, worsening DAI, colon length reduction, or in improving histological score. We found that our PBP8 CsgA-TFF3 is more effective in terms of reducing weight loss, DAI and increasing colon length. It is possible that the we observed little difference between wt-CsgA and CsgA-TFF3 in the cytokine panel analysis in our experiments because of length of the recovery period at the end of our experiment, and because the mechanism of TFF3 action is through mucosal healing and barrier function, not on reduction in cytokine levels. We have included additional text in the discussion section as follows (Line 465):

"It is worth nothing that when comparing our results with current literature reports on the use of traditional, oral anti-inflammatory drugs such as 5-aminosalicylic acid (5-ASA) for DSS colitis murine model treatment^{58-60(in text)}, we found that our PATCH technology seemed to improve the weight, DAI, and colon length of the mice to a better extent than 5-ASA. Future studies include side-by-side comparison and combination of PATCH and 5-ASA would be interesting to pursue as one of the multi-pronged approach toward IBD treatment."

7. Figures 4A and 6B: Different strains were compared. For clarity, I recommend explaining the difference observed in the strains compared.

The two figures represent different data sets. The purpose of Figure 4A was to demonstrate the residence time of the strains in the gut after a single oral administration, under conditions that were relevant for our study. We used the Prop-Luc strain of *E. coli* Nissle for this purpose in order to facilitate microbial tracking. The purpose of Figure 6B was to demonstrate the efficacy of the CsgA-TFF3 producing strain in a DSS model with rectal delivery. We used the PBP8 strain for those experiments because tracking was not as important in that context. There is a difference between the genotypes of the strains, in that Prop-Luc has the lux operon incorporated and has the native curli genes, whereas PBP8 has no lux operon and the native chromosomal genes have been deleted. There was no experiment in which Prop-Luc and PBP8 were compared against one another directly. However, we do not feel that this negatively affects the interpretation of the data in either case.

8. In this study, changes in the expression of IL-6, IL-17A and TNF-alpha were studied. I recommend studying the changes in expression of other cytokines, such as IL-4, IL-9 and IL-13, which are possibly regulated by the pathogenesis of IBD- or DSS-induced colitis.

We have some data on other cytokines already – IL-1 β , IL10 and IFN- γ were shown in the supplementary data. We did observe a reduction in IL-1 β between the wt-CsgA and CsgA-TFF3

conditions, which supports the conclusions drawn from the IL-6 and IL-17A data for Th17 response. However, it was not statistically significant, so we decided it was not important to include in the main text. As for other cytokines, we do not think they would be affected as much since the acute DSS model affects mostly innate immune responses and possibly some Th17 response due to pathogen invasion. Changes in cytokines such as IL-4, 9 and 13, which are directly related to Th2 response, might not be elicited as much in such a model¹⁴.

9. A recent study (<https://www.ncbi.nlm.nih.gov/pubmed/28627600>), microRNA-7-5p (miR-7-5p) was identified as a negative regulator of TFF3 and found to be upregulated in lesion tissues. Is a similar regulator involved in the regulation of TFF3 upon PATCH treatment?

Though we have not probed the amount of miR-7-5p in our experiments, we do have qRT-PCR data showing no changes in the amount of *tff3* mRNA expressed in the gut tissues among all the mice receiving DSS across all treatments (Figure S6-C). Therefore, the miR-7-5p regulator did not seem relevant to TFF3 levels upon PATCH treatment.

10. In this study, the plasmid with antibiotic resistance was used to express CsgA-TFF3. As mentioned in "section" discussion, a second generation PATCH system will be developed by using an antibiotic-free plasmid system. I recommend discussing the strategy of (1) expressing csgA-TFF3 upon induction by inducers which are more relevant to the microenvironment of IBD rather than arabinose, and (2) integrating the csgA-TFF3 cassette into the genome, which may overcome the potential issue on instability of the plasmid and CsgA-TFF3 expression.

We agree with the reviewer that these new induction methods would be interesting and relevant to our research. We have amended the main text as following to highlight their potential role in downstream work (Line):

"Otherwise, the PATCH system could potentially be integrated into the genome of the bacteria, though the low copy number of the genes might result in lower amount of therapeutics. Regarding the inducers, we can replace the arabinose inducible promoter with environmentally sensitive promoters that respond to temperature or inflammatory markers to avoid the use of external inducers and further improve the system."

Minor comments:

1. The title of the manuscript is not informative enough. I recommend providing a more descriptive title.

We have changed the title of the manuscript to "Engineered E. coli Nissle for the delivery of matrix-tethered therapeutic domains to the gut" so that it is more descriptive.

2. Figures 2C-G and 6F-J: For clarity, please provide labels or names e.g. GFP wt-CsgA in each image instead of using characters, e.g. C, D, etc.

The figures have been modified as suggested.

3. Figures 3: Please change "S. Typhi" in X-axis to italic "S. typhi", and adjust the colour scheme of the bars in Figure 3D to be consistent with those in Figure 3A-C. Furthermore, please make the font of "S. typhi" in X-axis consistent.

The figure has been modified as suggested.

4. Figure 6D: The exact p-values are recommended for the ns dataset.

The figure has been modified as suggested.

5. Line 205: CFU should be presented using CFU/g faecal sample.

The text has been modified as suggested.

6. Line 267: Please rephrase "over several days" to be more specific, e.g. "over five days".

The text has been modified as suggested.

7. In section Materials and Methods: Please change "Arabinose" to "arabinose".

The text has been modified as suggested.

8. Line 335: Please change "This could the explained..." to "This should be explained".

The text has been modified as suggested.

9. Line 338: Please change "form" to "from".

The text has been modified as suggested.

10. Line 541: The additional space should be removed in "3% ".

The text has been modified as suggested.

References

- 1 Monteiro, C. *et al.* Characterization of cellulose production in Escherichia coli Nissle 1917 and its biological consequences. *Environ Microbiol* **11**, 1105-1116, doi:10.1111/j.1462-2920.2008.01840.x (2009).
- 2 Kleta, S. *et al.* Role of F1C fimbriae, flagella, and secreted bacterial components in the inhibitory effect of probiotic Escherichia coli Nissle 1917 on atypical enteropathogenic E. coli infection. *Infect Immun* **82**, 1801-1812, doi:10.1128/IAI.01431-13 (2014).
- 3 Sonnenborn, U. & Schulze, J. The non-pathogenic Escherichia coli strain Nissle 1917 – features of a versatile probiotic. *Microbial Ecology in Health and Disease* **21**, 122-158, doi:10.3109/08910600903444267 (2009).
- 4 Fabrega, M. J. *et al.* Intestinal Anti-inflammatory Effects of Outer Membrane Vesicles from Escherichia coli Nissle 1917 in DSS-Experimental Colitis in Mice. *Front Microbiol* **8**, 1274, doi:10.3389/fmicb.2017.01274 (2017).
- 5 Chen, F. *et al.* Occludin is regulated by epidermal growth factor receptor activation in brain endothelial cells and brains of mice with acute liver failure. *Hepatology (Baltimore, Md.)* **53**, 1294-1305, doi:10.1002/hep.24161 (2011).
- 6 Martin, T. A. & Jiang, W. G. Loss of tight junction barrier function and its role in cancer metastasis. *Biochimica et Biophysica Acta (BBA) - Biomembranes* **1788**, 872-891, doi:<https://doi.org/10.1016/j.bbamem.2008.11.005> (2009).
- 7 Aamann, L., Vestergaard, E. M. & Grønbaek, H. Trefoil factors in inflammatory bowel disease. *World Journal of Gastroenterology : WJG* **20**, 3223-3230, doi:10.3748/wjg.v20.i12.3223 (2014).
- 8 Lin, N., Xu, L.-f. & Sun, M. The protective effect of trefoil factor 3 on the intestinal tight junction barrier is mediated by toll-like receptor 2 via a PI3K/Akt dependent mechanism. *Biochemical and Biophysical Research Communications* **440**, 143-149, doi:<https://doi.org/10.1016/j.bbrc.2013.09.049> (2013).
- 9 Vandembroucke, K. *et al.* Active delivery of trefoil factors by genetically modified Lactococcus lactis prevents and heals acute colitis in mice. *Gastroenterology* **127**, 502-513, doi:10.1053/j.gastro.2004.05.020 (2004).

- 10 Steidler, L. *et al.* Treatment of Murine Colitis by *Lactococcus lactis* Secreting Interleukin-10. *Science* **289**, 1352-1355, doi:10.1126/science.289.5483.1352 (2000).
- 11 Vandenbroucke, K. *et al.* Orally administered *L. lactis* secreting an anti-TNF Nanobody demonstrate efficacy in chronic colitis. *Mucosal Immunol* **3**, 49-56, doi:10.1038/mi.2009.116 (2010).
- 12 Jin, B.-R. *et al.* Rosmarinic acid suppresses colonic inflammation in dextran sulphate sodium (DSS)-induced mice via dual inhibition of NF- κ B and STAT3 activation. *Scientific reports* **7**, 46252-46252, doi:10.1038/srep46252 (2017).
- 13 Li, Y.-h. *et al.* Addition of Berberine to 5-Aminosalicylic Acid for Treatment of Dextran Sulfate Sodium-Induced Chronic Colitis in C57BL/6 Mice. *PLOS ONE* **10**, e0144101, doi:10.1371/journal.pone.0144101 (2015).
- 14 Alex, P. *et al.* Distinct cytokine patterns identified from multiplex profiles of murine DSS and TNBS-induced colitis. *Inflammatory bowel diseases* **15**, 341-352, doi:10.1002/ibd.20753 (2009).

Reviewers' Comments:

Reviewer #1:

Remarks to the Author:

All my previous critiques were addressed. This is a potentially important delivery mechanism and the synergistic effects of CsgA and TFFs would be a good area of further research. I would like to congratulate the authors for a nice study.

Reviewer #2:

Remarks to the Author:

In its current form, it is acknowledged that neutralizing antibodies could be a problem for any protein-based therapeutic. The authors failed to address the major concern related to the need for providing some mechanistic insights (such as on wound healing) and for demonstrating the efficacy with appropriate controls (such as CsgA or TFF alone). Consequently, this reviewer is not convinced on the robustness of the anti-inflammatory effect of the engineered strain in mice that are spontaneously prone to develop colitis (such as IL-10 deficient mice). Alternatively, a chronic model of DSS-induced colitis shall be employed.

Reviewer #3:

Remarks to the Author:

In the revised manuscript, the authors have sufficiently addressed most of the comments made previously. Therefore, the quality of the manuscript has significantly improved. However, a few points should be further addressed before accepting this work for publishing in Nature Communications.

1. Figure 6: A positive control (e.g. 5-ASA) that can ameliorate inflammation in the DSS-induced IBD animal model should be included in the assays carried out in this work. The results obtained would help the authors to further determine the performance of engineered EcN strain versus a positive control.
2. Figures 5, 6 and 7: The authors investigated the resistance conferred by the probiotic strain expressing CsgA-TFF3. The EcN strain expressing TFF3 alone should be included as a control to sufficiently support the conclusion made in this work. Although the authors argued that the inclusion of the TFF3-expressing strain is beyond the scope of this revision, such experimental data obtained from the TFF3 expression is critical and necessary, and will add value to this work.

Responses to the editors are in blue

Texts from the previous iteration of the manuscript are in green and italics

Newly added texts are in blue, italics and highlighted

Response to the editors

Your manuscript entitled "Probiotic Associated Therapeutic Curli Hybrids (PATCH)" has now been seen by 3 referees. You will see from their comments below that while they find your work of interest, some important points are raised. We are interested in the possibility of publishing your study in Nature Communications, but would like to consider your response to these concerns in the form of a revised manuscript before we make a final decision on publication.

We therefore invite you to revise and resubmit your manuscript, taking into account the points raised. Please highlight all changes in the manuscript text file.

After discussing the concerns of the reviewers with my colleagues, we ask that the final set of requested controls be added to the manuscript.

Based on the most recent round of reviewer comments, two additional conditions were requested as controls for the mouse experiments: 1) rectal administration of an EcN strain engineered to secrete only the soluble form of the TFF3 domain, and 2) oral administration of 5-ASA as a positive control for colitis treatment. Of these, Reviewer #3's comments place more importance on the soluble TFF3 condition, presumably to identify the difference in protective effect between soluble and CsgA-bound TFF3.

Therefore, we engineered a strain of EcN (also called PBP8 in the manuscript) that secretes TFF3 into the extracellular medium as a soluble entity. This was accomplished by fusing the Sec and N22 secretion tags directly to the N-terminus of the TFF3 sequence, without the intervening CsgA domain. We performed another trial of the murine DSS colitis study, using identical conditions as we used in the original manuscript Figure 6, but with only three conditions: DSS only, DSS plus PBP8 secreting soluble TFF3, and DSS plus PBP8 secreting CsgA-TFF3. During this trial, none of the mice exhibited weight loss or DAI scores as severe as the mice in our original set of experiments. It is known that the severity of the acute DSS model can be highly variable based on several factors that are difficult to control for. Although we thought the severity of these disease markers could have increased if we had extended the injury period, we opted not to do this in order to keep the protocol identical to the original trial. The mean histology score from the CsgA-TFF3 group was the lowest of the three groups, but the difference was not statistically significant.

However, we contend that the new data do not contradict our previous work, and have included them in the supplementary materials (Figure S6). Indeed, we assert in the paper that CsgA fusion of therapeutic domains is a complementary strategy to the release of soluble therapeutic factors, and that the two modes of delivery could work in concert.

While we understand and appreciate the values of a positive control, we also would like to return a revision draft to the editors in a timely manner. We found that a positive control such as 5-ASA would require us to submit a protocol revision to the animal care and use committee at Harvard which would result in several months of additional delay before we could start another mouse experiment, which can also be lengthy already in its nature. Moreover, based on the literature about 5-ASA that we have mentioned in the latest draft of our manuscript (shown below), we do not believe that 5-ASA is as potent in terms of the treatment of acute DSS-induced colitis as to serve as an appropriate positive control.

“It is worth nothing that when comparing our results with current literature on the use of traditional, oral anti-inflammatory drugs such as 5-aminosalicylic acid (5-ASA) for DSS colitis murine model treatment⁶⁰⁻⁶², we found that our PATCH technology seemed to improve the weight, DAI, and colon length of the mice to a better extent than 5-ASA. Future studies include side-by-side comparison and combination of PATCH and 5-ASA would be interesting to pursue as a multi-pronged approach toward alleviating colitis.”

This might result in more optimization in order to find an appropriate positive control, leading to a delay in the publication process and might not yield a productive outcome. Reviewer #3's comments seem to put more weight on the second type of control, the EcN strain expressing TFF3 alone. Therefore, we do not think it would be productive to include a 5-ASA condition in the mouse experiment.

We did manage to include the second type of control, which is PBP8 strain expressing TFF3 without CsgA as requested. We have cloned another construct that only contains the N22 signaling peptide of CsgA and TFF3 without the structural unit of CsgA. Hence, the N22-TFF3 would be expressed and secreted with the exact mechanisms that CsgA-TFF3 utilized, but it would not form amyloid fibers due to a lack of the CsgA structure. We transformed the N22-TFF3 construct to PBP8 and chose to perform the *in vivo* acute DSS-induced intestinal injury experiment similar to that presented in figure 6. The explanation of this new experiment can be found in the newly added text below and in the revised manuscript:

“In order to assess the effects of CsgA fusion on the bioactivity of TFF3, we performed a trial experiment with a strain of PBP8 engineered to secrete TFF3 in a soluble form. PBP8 was transformed with pBbB8k-N22-TFF3, a plasmid encoding the N22 secretion signal peptide followed by the TFF3 encoding sequence. Thus, the new PBP8 N22-TFF3 strain secreted TFF3 in a soluble form through the same curli secretion machinery as CsgA-TFF3. Using identical protocol to the original DSS induced colitis experiment (Figure 6), we ran a smaller experiment with mice randomly assigned to one of three experimental groups: colitic group (PBS DSS⁺), DSS treated soluble TFF3 group (PBP8 N22-TFF3 DSS⁺) and DSS treated curli-bound TFF3 group (PBP8 CsgA-TFF3 DSS⁺). The results of this experiment mirrored those of the original – CsgA-TFF3 continued to receive the lowest histology scores, though the difference between the groups was not statistically significant based on our metrics (Figure S6).

This is likely due to high variability in the DSS model, since the mice in the colitic group did not become injured as severely as they did in the original experiment, despite the identical protocol.”

“Ongoing work in our lab is focused on probing the difference between the secretion of various curli-tethered (via fusion to CsgA) and soluble therapeutic proteins. Although we did not observe a difference between tethered and untethered TFF3 in this work, the multivalency and mucoadhesion offered by the CsgA-TFF3 scaffold could be advantageous for increasing the local concentration of drugs in the gut. Indeed, future iterations of the PATCH platform may allow for synergy between tethered and soluble therapeutic domains.”

We noticed that reviewer #3 also like to see a similar fluorescence staining experiment in figure 5 with this new control. However, we felt that the experiments in figure 6 were more crucial to the overall narrative of the manuscript. Therefore, to achieve a reasonable turnaround time of this manuscript to the editors, we decided to perform the experiments in figure 6 with the new control and leave the experiments in figure 5 for the future investigation.

However, we do not require the addition of further mouse models as indicated by Reviewer #2. In the absence of these models, we ask that claims of application to colitis are toned down and claims of application to IBD are removed. We ask that the manuscript focus on the proof-of-principle work using the PATCH system to treat acute DSS-induced inflammation. Elaboration about potential future directions into a chronic model are welcome in the Discussion section.

In the most recent iteration of the manuscript, we have already tuned down our claims in terms of the general application to colitis. We made sure that we did not claim the utilization of PATCH system for the IBD treatment. Rather, we focused more on a specific mechanism, which is a mucosal healing. We have included texts such as:

“During IBD disease flare-ups, the barrier formed by the GI epithelium is disrupted, exposing the gut lining to potentially injurious agents like bacteria and their products or extreme pH⁶. This is true for other chronic inflammatory symptoms like ulcers or fistula as well. Therefore, the speedy restoration of mucosal healing and epithelial integrity is essential for treating such symptoms. The epithelial mucosa heal through restitution and regeneration processes. Complete regeneration is a slower process that relies on stem cell proliferation and differentiation. In contrast, restitution can occur within hours after injury and relies on the migration of epithelial cells from the surrounding area into the wound site. This process can restore mucosal continuity to the gut lining and protect it from bacteria and foreign antigens, and fluid and electrolyte losses, which prevent further inflammatory processes⁷. Although epithelial restitution is essential in protecting the GI tract during insult, it is difficult to monitor as an outcome

directly in clinical studies⁸. Most therapeutics for IBD and ulcers focus on modulating inflammatory pathways, leaving room for therapeutic advancement in mucosal healing.”

“Here we present an alternative approach to engineered microbial therapies to promote mucosal healing.”

We have included more text in the introduction to further emphasize that this study provides a proof-of-concept work using PATCH to treat acute DSS-induced inflammation, as shown below:

“We demonstrate that PATCH is capable of ameliorating inflammation caused by dextran sodium sulfate (DSS) induced colitis in a mouse model.”

In this iteration of the manuscript, we elaborated more about the potential future experiments related to chronic colitis models such as chronic DSS-induced colitis, il10 knockout and adoptive T-cell transfer, in the discussion section as following:

“We chose the acute DSS induced colitis model because of its practical accessibility and its appropriateness for studying mucosal healing in the mammalian GI tract. However, the potential applicability of PATCH for the treatment of diseases like IBD will require further studies in complementary disease model systems (e.g. IL-10 knockout, adoptive T-cell transfer, TNBS injury).”

We are committed to providing a fair and constructive peer-review process. Do not hesitate to contact us if you wish to discuss the revision in more detail or if there are specific requests from the reviewers that you believe are technically impossible or unlikely to yield a meaningful outcome.

Reviewers' comments (For reference):

Reviewer #1 (Remarks to the Author):

All my previous critiques were addressed. This is a potentially important delivery mechanism and the synergistic effects of CsgA and TFFs would be a good area of further research. I would like to congratulate the authors for a nice study.

Reviewer #2 (Remarks to the Author):

In its current form, it is acknowledged that neutralizing antibodies could be a problem for any protein-based therapeutic. The authors failed to address the major concern related to the need for providing some mechanistic insights (such as on wound healing) and for demonstrating the efficacy with appropriate controls (such as CsgA or TFF alone). Consequently, this reviewer is not convinced on the robustness of the anti-inflammatory effect of the engineered strain in mice that are spontaneously prone to develop colitis (such as IL-10 deficient mice). Alternatively, a chronic model of DSS-induced colitis

shall be employed.

Reviewer #3 (Remarks to the Author):

In the revised manuscript, the authors have sufficiently addressed most of the comments made previously. Therefore, the quality of the manuscript has significantly improved. However, a few points should be further addressed before accepting this work for publishing in Nature Communications.

1. Figure 6: A positive control (e.g. 5-ASA) that can ameliorate inflammation in the DSS-induced IBD animal model should be included in the assays carried out in this work. The results obtained would help the authors to further determine the performance of engineered EcN strain versus a positive control.
2. Figures 5, 6 and 7: The authors investigated the resistance conferred by the probiotic strain expressing CsgA-TFF3. The EcN strain expressing TFF3 alone should be included as a control to sufficiently support the conclusion made in this work. Although the authors argued that the inclusion of the TFF3-expressing strain is beyond the scope of this revision, such experimental data obtained from the TFF3 expression is critical and necessary, and will add value to this work.

Reviewers' Comments:

Reviewer #3:

Remarks to the Author:

In the revised manuscript, the authors have sufficiently addressed the comments made previously.